

# Heuristic Estimation of Low-Level Cloud Fraction over the Globe Based on a Decoupling Parameterization

Sungsu Park[1] and Jihoon Shin[1]

[1]School of Earth and Environmental Sciences, Seoul National University, Seoul, South Korea

**Correspondence:** Sungsu Park (sungsup@snu.ac.kr)

**Abstract.** Based on the decoupling parameterization of the cloud-topped planetary boundary layer, a simple equation is derived to compute the inversion height. In combination with the lifting condensation level and the amount of water vapor in near-surface air, we propose a low-level cloud suppression parameter (LCS) and estimated low-level cloud fraction (ELF), as new proxies for the analysis of the spatiotemporal variation of the global low-level cloud amount (LCA). Individual surface and
5 upper-air observations are used to compute LCS and ELF as well as lower-tropospheric stability (LTS), estimated inversion strength (EIS), and estimated cloud-top entrainment index (ECTEI), three proxies for LCA that have been widely used in previous studies. The spatiotemporal correlations between these proxies and surface-observed LCA were analyzed.

Over the subtropical marine stratocumulus deck, both LTS and EIS well diagnose seasonal-interannual variations of LCA. However, their use as global proxy for LCA is limited due to their weaker and inconsistent relationship with LCA over land.
EIS is anti-correlated with the decoupling strength more strongly than it is correlated with the inversion strength. Compared with LTS and EIS, ELF and LCS better diagnose temporal variations of LCA, not only over the marine stratocumulus deck but also in other regions. However, all proxies have a weakness in diagnosing interannual variations of LCA in several subtropical stratocumulus decks. In the analysis using all data, ELF achieves the best performance in diagnosing spatiotemporal variation of LCA, explaining about 60% of the spatial-seasonal-interannnual variance of the seasonal LCA over the globe, which is a
much larger percentage than those explained by LTS (2%) and EIS (4%).

Our study implies that accurate prediction of inversion base height and lifting condensation level is a key factor necessary for successful simulation of global low-level clouds in general circulation models (GCMs). Strong spatiotemporal correlation between ELF (or LTS) and LCA identified in our study can be used to evaluate the performance of GCMs, identify the source of inaccurate simulation of LCA, and better understand climate sensitivity.

*Copyright statement.*

## 1 Introduction

Clouds belong to the most important but uncertain components of the climate system. Due to their strong shortwave radiative cooling effect on the Earth, low-level clouds have been the focus of various studies in the past few decades, both in the





observation and modeling communities. Slingo (1990) estimated that a 4% increase of the low-level cloud amount (LCA) has

the potential to offset global warming associated with a doubled $CO_2$ concentration. Most low-level clouds exist over the ocean,

mainly due to abundant moisture sources near the surface (Hahn and Warren, 1999). Among various types of low-level clouds,

marine stratocumulus clouds (MSC) have received special attention due to their large spatial coverage and the complexity of

physics and dynamic processes controlling their formation and dissipation (Wood, 2012). Several planetary boundary layer

(PBL) schemes used in general circulation models (GCM) have the capability to simulate MSC and associated feedback

processes in a realistic way (e.g., Lock et al. (2000), Bretherton and Park (2009), Park and Bretherton (2009)). However, MSC

simulated by these parameterization schemes or more complex numerical models are the results of complex interactions among

various physics processes. Therefore, it has not been easy for the climate researchers to understand the feedback processes of

MSC from the climate perspective. If a simple proxy can diagnose spatial and temporal variations of MSC, it would be much

easier for the climate researcher to understand the role of MSC in future climate, both qualitatively and quantitively.

    Klein and Hartmann (1993) (KH93 hereafter) showed that a lower tropospheric stability, LTS $\equiv \theta_{700} - \theta_{1000}$ where $\theta_{700}$

and $\theta_{1000}$ are the potential temperatures at 700 and 1000 [hPa] levels, respectively, correlates well with seasonal variations of

LCA over the subtropical marine stratocumulus deck. LTS has been widely used as a proxy to understand the characteristics

of MSC and their impact on future climate. The success of LTS stems in part from the fact that LTS is correlated with other

factors controlling the formation of MSC in the subtropical marine stratocumulus deck such as the sea surface temperature

(SST), cold air advection, free tropospheric moisture, and subsidence in association with a subtropical high-pressure system.

Using a heuristic Lagrangian MSC model, Park et al. (2004) (PLR04 hereafter) explored the sensitivity of MSC to various

environmental conditions in the cold advection regime of the northeastern subtropical Pacific and to both warm and cold

advection regimes of the eastern equatorial Pacific Ocean. Consistent with Klein (1997), PLR04 found a positive correlation

between the simulated MSC fraction and strong upstream subsidence, although Myers and Norris (2013) reported the opposite

correlation. PLR04 simulated less MSC with a drier free atmosphere. However, enhanced longwave radiative cooling at the

top of MSC capped by a drier free atmosphere (this process was not included in the PLR04's model) may increase MSC by

enhancing turbulent vertical moisture transport from the sea surface to overlying MSC.

Based on the decoupling hypothesis suggested by PLR04 and other proceeding works [e.g., Augstein et al. (1974), Albrecht

et al. (1979), Betts and Ridgway (1988), and Bretherton (1992)], Wood and Bretherton (2006) (WB06 hereafter) extended

KH93's LTS and suggested an estimated inversion strength (EIS) as better proxy for LCA in which temperature profiles in the

decoupled layer below the inversion and the free troposphere above the inversion are assumed to be close to a moist adiabat

that is strongly temperature-dependent. WB06 showed that compared with LTS, EIS correlates better with LCA over a wide

range of stratocumulus regimes, because it captures the lapse rate and boundary layer structure more completely than LTS.

Similar to LTS, EIS has been widely used as a good proxy for LCA.

    In their derivation of EIS, WB06 assumed that the factor $z_{inv} \cdot (\Gamma_{700}^m - \Gamma_{LCL}^m)$, where $z_{inv}$ is the inversion height and $\Gamma_{700}^m$

and $\Gamma_{LCL}^m$ are the moist adiabatic lapse rates at 700 [hPa] and lifting condensation level (LCL) of near-surface air ($z_{LCL}$),

respectively, contributes less to the correlation relationship between the inversion strength and LTS than the other terms, so

that it was simply set to zero. Although a scaling argument was provided to justify this simplification, a more fundamental



reason for neglecting this term was associated with the practical difficulty in estimating and parameterizing $z_{inv}$. PLR04 suggested a heuristic parameterization for PBL decoupling to simulate stratocumulus cloudiness in the inversion-capped marine boundary layer. In their study, PBL decoupling is parameterized as an increasing function of the height difference between $z_{inv}$ and $z_{LCL}$. PLR04's conceptual idea of PBL decoupling was used in the WB06's derivation of EIS and other studies to

5 understand the variation of cloudiness associated with PBL decoupling (Stevens (2006); Xiao et al. (2011); Van der Dussen et al. (2014); Park (2014a); Park (2014b); Dal Gesso et al. (2015a); Dal Gesso et al. (2015b); Van der Dussen et al. (2015); Van der Dussen et al. (2016); Neggers et al. (2017)). In our study, we suggest a simple heuristic method to estimate $z_{inv}$ by combining PLR04's decoupling parameterization with EIS. By using $z_{inv}$, $z_{LCL}$, and water vapor specific humidity in the surface-based mixed layer ($q_{v,ML}$), we propose two low-level cloud suppression parameters (LCS), $\beta_1 \equiv (z_{inv} + z_{LCL})/\Delta z_s$

and $\beta_2 \equiv \sqrt{z_{inv} \cdot z_{LCL}}/\Delta z_s$ with $\Delta z_s = 2750$ [m], and an estimated low-level cloud fraction, ELF$\equiv f(1-\beta_2)$ with $f \equiv max[0.15, min(1, q_{v,ML}/0.003)]$, as new proxy for the characterization of the spatiotemporal variation of LCA over the globe. Individual surface and upper-air observations are used to compute LTS, EIS, LCS, and ELF, and the correlations between these proxies and the surface-observed seasonal LCA are examined. We also analyzed the recently proposed estimated cloud-top entrainment index [ECTEI, Kawai et al. (2017)], which is a modified EIS that takes into account a cloud-top entrainment

criteria. It will be shown that compared with LTS, EIS, and ECTEI, which are mainly designed as proxies for marine LCA, ELF and LCS are better proxies for the global LCA, applicable over both the ocean and land.

The structure of this paper is as follows. Section 2a provides a detailed explanation on the conceptual framework used to compute $z_{inv}$, LCS, ELF, and other related proxies for LCA. Section 2b describes the data and analysis method. The correlations between various proxies and LCA in spatial and temporal domains over the globe are presented in Section 3. A

20 summary and conclusion are provided in Section 4.

## 2 Method

### 2.1 Conceptual Framework

Following PLR04 and WB06, we assume that the lower troposphere below 700 hPa consists of four regimes (see Fig 1): a surface-based mixed layer (ML) topped at $z_{ML}$ with the potential temperature, $\theta_{ML}$ and water vapor specific humidity, $q_{v,ML}$,

specified at the reference height, $z_{ref}$ or $p_{ref}$ (i.e., $\theta_{ML} = \theta_{ref}$, $q_{v,ML} = q_{v,ref}$); a decoupled cloud layer (DL) with a vertical gradient of $\theta$ approximated by the moist $\theta$ adiabat at $z_{ML}$ ($\Gamma_{DL}^m > 0$); an inversion at the DL top ($z_{inv}$); and the free atmosphere with a vertical gradient of $\theta$ approximated by the moist $\theta$ adiabat at $p = 700$ hPa ($\Gamma_{700}^m > 0$). The moist adiabatic lapse rate of $\theta$ used in our study ($\Gamma_{DL}^m$ and $\Gamma_{700}^m$ in unit of $[K \cdot m^{-1}]$) is a function of $T$ and $p$; it increases as $p$ increases or $T$ increases. The inversion strength at $z_{inv}$, IS $\equiv \theta_{inv}^+ - \theta_{inv}^-$ becomes

$$IS = LTS + \Gamma_{DL}^m \cdot z_{ML} - \Gamma_{700}^m \cdot z_{700} + z_{inv} \cdot (\Gamma_{700}^m - \Gamma_{DL}^m), \tag{1}$$

where $\theta_{inv}^+$ is the potential temperature just above the inversion, $\theta_{inv}^-$ is the potential temperature just below the inversion, and LTS $\equiv \theta_{700} - \theta_{ML}$ is the lower tropospheric stability. By assuming that the contribution of $z_{inv} \cdot (\Gamma_{700}^m - \Gamma_{DL}^m)$ to variability




in the relationship between IS and LTS is negligibly small due to the opposite variations of $z_{inv}$ and $\Gamma^m_{700} - \Gamma^m_{DL}$ with LTS, WB06 derived the so-called estimated inversion strength, EIS,

$$EIS = LTS + \Gamma^m_{DL} \cdot z_{ML} - \Gamma^m_{700} \cdot z_{700}, \tag{2}$$

which was shown to be a better proxy for LCA than LTS. Because it does not contain $z_{inv}$, which is hard to estimate, EIS has
been used as a convenient proxy for LCA.

If $z_{inv}$ can be reasonably estimated instead of being neglected, it may be possible to construct a better proxy than EIS for LCA. In this study, we suggest an approach to estimate $z_{inv}$ and other related proxies in a heuristic way based on the decoupling hypothesis suggested by PLR04. From the analysis of sounding data, PLR04 showed that the decoupling parameter $\alpha$ can be parameterized as an increasing function of the decoupled layer thickness, $\Delta z_{DL} \equiv z_{inv} - z_{ML}$,

$$\alpha \equiv \frac{\theta^-_{inv} - \theta_{ML}}{\theta^+_{inv} - \theta_{ML}} \approx \left( \frac{\Delta z_{DL}}{\Delta z_s} \right)^\gamma \approx \left( \frac{z_{inv} - z_{ML}}{\Delta z_s} \right), \tag{3}$$

where the scale height, $\Delta z_s \approx 2750 \, [m]$ is obtained from the analysis of a set of sounding data over the ocean (see PLR04). Originally, PLR04 defined $\alpha$ for the condensate potential temperature $\theta_c \equiv \theta - (L_v/C_p) \cdot q_l - (L_s/C_p) \cdot q_i$ with $\gamma \approx 1.1 - 1.3$, where $q_l$ and $q_i$ are the cloud liquid and ice contents, respectively, and $\theta_c$ is a conserved scalar with respect to the phase change. In our study, however, $\alpha$ is defined for $\theta$ and accordingly, $\gamma$ is slightly reduced to 1 to account for $\alpha_\theta > \alpha_{\theta_c}$ in the cloud-topped
decoupled layer. The choice of $\gamma = 1$ allows us to obtain analytical expressions for various proxies as will be shown below. The small (large) $\alpha$ indicates that the environmental air at the inversion base, $z^-_{inv}$ is well connected to (decoupled from) the surface air property with abundant moisture, providing more (less) favorable conditions for the formation of LCA at $z^-_{inv}$. It should be noted that $\alpha$ only measures the degree of vertical decoupling of thermodynamic properties between $z_{ML}$ and $z^-_{inv}$. That is, $\alpha$ does not provide information regarding the amount of surface moisture.
By combining Eq.(3) with $\theta^+_{inv} = \theta_{700} - \Gamma^m_{700} \cdot (z_{700} - z_{inv})$ and $\theta^-_{inv} = \theta_{ML} + \Gamma^m_{DL} \cdot (z_{inv} - z_{ML})$, we can derive the following expressions for the inversion height,

$$z_{inv} = -(LTS/\Gamma^m_{700}) + z_{700} \qquad\qquad + \Delta z_s \cdot \left( \frac{\Gamma^m_{DL}}{\Gamma^m_{700}} \right)$$

$$= -(EIS/\Gamma^m_{700}) + z_{ML} \cdot \left( \frac{\Gamma^m_{DL}}{\Gamma^m_{700}} \right) + \Delta z_s \cdot \left( \frac{\Gamma^m_{DL}}{\Gamma^m_{700}} \right), \tag{4}$$

which can be also written as $z_{inv} = \alpha \cdot \Delta z_s + z_{ML}$ from Eqn.(3). Then, the inversion strength, IS becomes

$$IS = (1 - \alpha) \cdot \Gamma^m_{DL} \cdot \Delta z_s, \tag{5}$$

and decoupling strength, DS $\equiv \theta^-_{inv} - \theta_{ML}$ becomes

$$DS = \alpha \cdot \Gamma^m_{DL} \cdot \Delta z_s, \tag{6}$$

and LTS = IS + DS + $\Gamma^m_{700} \cdot (z_{700} - z_{inv})$, as shown in Figure 1. Once $\theta_{ML} = \theta_{ref}$, $q_{v,ML} = q_{v,ref}$, and $z_{ML}$ are obtained, we can consecutively compute LTS = $\theta_{700} - \theta_{ref}$ and $z_{inv}$ using the first expression of Eq.(4), $\alpha$ from Eq.(3), IS from Eq.(5),





and DS from Eq.(6). Note that IS is identical to the sum of EIS and $z_{inv} \cdot (\Gamma_{700}^m - \Gamma_{DL}^m)$, the neglected term in the original formulation of WB06.

Following previous studies (e.g., PLR04 and WB06), we will assume that $z_{ML} \approx z_{LCL}$ over the ocean. However, due to insufficient moisture at the surface, it is likely that $z_{ML} < z_{LCL}$ over most land areas unless strong buoyancy or shear production near the surface sufficiently deepens the surface-based mixed layer. It may be possible to parameterize $z_{ML}$ as a function of turbulent kinetic energy within the PBL; however, for simplicity, we assume that $z_{ML} \approx z_{LCL}$ over the entire globe.

As mentioned above, $\alpha$ only measures the degree of vertical decoupling of thermodynamic properties between the inversion base and surface air, not the surface moisture itself. Conceptually, however, the formation of LCA at the inversion base is likely to be influenced by both surface moisture and $\alpha$. As a simple but practical proxy representing surface moisture, we select $z_{LCL}$. From the simple conceptual argument that small (large) $z_{LCL}$ and $\alpha$ are likely to be associated with large (small) LCA at the inversion base, we define the first low-level cloud suppression parameter (LCS), $\beta_1$, as

$$\beta_1 \equiv \alpha + \mu \cdot \left( \frac{z_{LCL}}{\Delta z_s} \right) = \frac{z_{inv} + z_{LCL}}{\Delta z_s}, \tag{7}$$

where the second equality is obtained by assuming $z_{ML} \approx z_{LCL}$ and $\mu = 2$. In principle, $\mu$ can be estimated in an empirical way using multiple linear regression analysis of LCA fitted to $z_{inv}$ and $z_{LCL}$ such that $\beta_1$ explains the maximum fraction of the variance of LCA. We performed a multiple linear regression analysis of LCA on $z_{inv}$ and $z_{LCL}$ using individual seasonal data in each $2.5^o$latitude x $5^o$longitude grid box over the globe. Except over some portions, the values of $\mu$ were mostly between 0 and 4 with an approximate average value, $\mu = 2$ (not shown). The resulting $\beta_1$ is a non-dimensional sum of $z_{inv}$ and $z_{LCL}$. By simply extending $\beta_1$, we also define the second LCS, $\beta_2$, as a non-dimensional product of $z_{inv}$ and $z_{LCL}$,

$$\beta_2 = \frac{\sqrt{z_{inv} \cdot z_{LCL}}}{\Delta z_s}, \tag{8}$$

which, similar to $\beta_1$, increases as the surface air becomes drier and the PBL deepens. In the case of $z_{ML} \approx z_{LCL}$, it becomes $\beta_1 \approx (z_{LCL}/\Delta z_s) \cdot [2 + (z_{inv} - z_{ML})/z_{ML}]$ and $\beta_2 \approx (z_{LCL}/\Delta z_s) \cdot \sqrt{1 + (z_{inv} - z_{ML})/z_{ML}}$, where the first factor in both formulae represents the degree of subsaturation of near-surface air, and the second factor represents the decoupling strength. Note that $z_{LCL}$ oppositely controls the surface moisture parameter and decoupling strength: small $z_{LCL}$ decreases the first factor but increases the second factor and vice versa. Large (small) values of $\beta_1$ and $\beta_2$ likely favor the dissipation (formation) of LCA at the inversion base.

Finally, we define the estimated low-level cloud fraction (ELF), as

$$ELF = f \cdot (1 - \beta_2) = f \cdot \left[ 1 - \frac{\sqrt{z_{inv} \cdot z_{LCL}}}{\Delta z_s} \right] \leq 1, \tag{9}$$

where $f$ is the freezedry factor (Vavrus and Waliser, 2008),

$$f = max \left[ 0.15, \, min \left( 1, \frac{q_{v,ML}}{0.003} \right) \right], \tag{10}$$





designed to reduce the parameterized cloud fraction in the extremely cold and dry atmospheric conditions typical of polar and high latitude winter. Cloud fraction parameterization based on the grid-mean relative humidity (RH) assumes that there is a certain amount of subgrid variability of thermodynamic scalars (e.g., Park et al. (2014)), which allows the formation of cloud fraction even when the grid-mean RH is smaller than 1. In the very stable Arctic and high-latitude atmosphere

during winter, however, there is little subgrid variability (Jones et al., 2004). Thus, any grid-mean RH-based cloud fraction parameterization (e.g., our LCS based on $z_{LCL}$) is likely to predict too much LCA there. Vavrus and Waliser (2008) showed that the implementation of the freezedry factor into GCM substantially reduced the simulated LCA in the Arctic and high-latitude regions during winter and improved the simulation. Compared to $\beta_2$, ELF diagnoses smaller LCA in the high-latitude region during winter where the amount of water vapor within the ML is often smaller than 3 [g kg$^{-1}$] (see Figs 2e,f). If

$z_{ML} \approx z_{LCL}$, it becomes ELF$= f \cdot [\, 1 - (z_{LCL}/\Delta z_s)\sqrt{1 + (z_{inv} - z_{ML})/z_{ML}}\,]$, where $f$ denotes the amount of water vapor in the surface-based ML air, $z_{LCL}$ represents the degree of subsaturation of near-surface air, and $(z_{inv} - z_{ML})/z_{ML}$ quantifies the degree of thermodynamic decoupling of the inversion base air from the surface-based ML air. Our ELF predicts that LCA increases as the near-surface air becomes more saturated with enough amount of water vapor and as the PBL becomes more vertically coupled. When the inversion base air is fully coupled with saturated near-surface air containing enough water vapor,

ELF approaches its upper bound of 1. Considering that low-level clouds usually form at $z_{inv}$, the conceptual cloud formation processes embedded in ELF is consistent with what is expected to happen in nature. We defined ELF using $\beta_2$ instead of $\beta_1$, because $\beta_2$ has a better global performance than $\beta_1$ (see Tables 1, 3, 4, 6). Note that our ELF can have small negative values, which, however, can be reset to zero for more complete parameterization. In our study, we don't reset ELF to zero.

In reality, the key factor controlling the formation of clouds is the relative humidity. According to our conceptual framework,

most of the clouds are likely to form at the inversion base. To compute the relative humidity at the inversion base, $RH^-_{inv}$, we follow PLR04 and assume that the decoupling parameter, $\alpha$ defined in Eq.(3) also describes the decoupling of the water vapor specific humidity, $q_v$. Then, the water vapor specific humidity ($q^-_{v,inv}$) and potential temperature ($\theta^-_{inv}$) at the inversion base can be computed as $q^-_{v,inv} = \alpha \cdot q^+_{v,inv} + (1-\alpha) \cdot q_{v,ML}$ and $\theta^-_{inv} = \alpha \cdot \theta^+_{inv} + (1-\alpha) \cdot \theta_{ML}$. The lapse rate of $q_v$ in the free atmosphere $\Gamma^{q_v}_{700}$ which is required to compute $q^+_{v,inv} = q_{v,700} - \Gamma^{q_v}_{700} \cdot (z_{700} - z_{inv})$, is obtained from the linear slope of $q_v$

at the 700 and 750 hPa levels. Using $q^-_{v,inv}$ and $\theta^-_{inv}$, we compute $RH^-_{inv}$ as an additional proxy for LCA. We note that only $RH^-_{inv}$ needs the information on $q^-_{v,inv}$; the other proxies do not need information on the vertical profile of $q_v$ other than $q_{v,ML}$ to compute $z_{LCL}$.

## 2.2 Data and Analysis

The surface observation data used in our study are from the Extended Edited Cloud Report Archive (EECRA, Hahn and

Warren (1999)) for January 1956 to December 2008 over the ocean (without sea ice) and January 1971 to December 1996 over land. The EECRA compiles individual ship and land observations of clouds (e.g., cloud amount and cloud type for each low-, middle-, and upper-level), present weather (e.g., fog, rain, snow, thunderstorm shower, and drizzle), and other coincident surface meteorologies (e.g., sea level pressure, sea surface temperature, ship-deck air temperature, dew-point depression, and wind speed and direction) at every 3 or 6 hours based on the strict hierarchy of the World Meteorological Organization (WMO,





WMO (1975)). Following Park and Leovy (2004), we filtered out the observations obtained under poor illumination conditions (i.e., the moonlight screening criteria, Hahn et al. (1995)), from the WMO's Historical Sea Surface Temperature (HSST) Data Project (identified by card deck numbers 150-156), or with any missing surface meteorologies or cloud information. In this study, we focus on the analysis of LCA of all cloud types including cumulus as well as stratus. The upper-level meteorologies

(e.g., $p$, $\theta$, and $q_v$) are from the ERA interim reanalysis products (ERAI, Simmons et al. (2007)) from January 1979 to December 2008 at 6-hourly time intervals. Spatial and temporal interpolations are performed to compute the upper-level meteorologies at the exact time and location at which the EECRA surface observers reported LCA. Because both EECRA and ERAI are necessary, our analysis only uses the data from January 1979 to December 2008 (30 years) over the ocean and January 1979 to December 1996 over land (18 years). Using the 6-hourly ERAI vertical profiles of $\theta$ and $q_v$ interpolated to individual EECRA

surface observations, we computed the twelve proxies for LCA (LTS, EIS, ECTEI, IS, DS, $\alpha$, $z_{LCL}$, $z_{inv}$, $RH_{inv}^-$, $\beta_1$, $\beta_2$, ELF) and averaged them into $5^o$latitude x $10^o$longitude seasonal data for each year. Thus, if there is no missing grid value, a total of 120 and 72 seasonal datapoints is available in each grid box over the ocean and land, respectively. To reduce the impact of random noise in association with the small observation number, the year with an observation number smaller than 10 in each season and grid box is not used in our analysis.

Three separate correlation analyses were performed between LCA and the twelve proxies: spatial-seasonal correlation analysis using climatological seasonal (i.e., DJF, MAM, JJA, SON) grid data (Table 1 and Figs 3-6; Table 4 and Figs 11,12) or climatological seasonal data averaged over selected regions (Figs 7,8; Fig 12), seasonal-interannual and interannual correlation analysis using seasonal grid data from each year (Figs 9-10; Fig 13) or seasonal data averaged over selected regions from each year (Table 2; Table 5), and combined spatial-seasonal-interannual correlation analysis (Table 3; Table 6) using seasonal grid

data over the globe in each year. For the spatial-seasonal correlation analysis, only the seasons and grid boxes with a climatological seasonal observation number equal to or larger than 100 are used. For any temporal correlation analysis containing interannual variations, we only used the years and seasons with a seasonal observation number equal or larger than 10 for each year within the grid box or selected regions. Only the grid boxes with the number of effective seasonal data points equal to or larger than 50 out of the maximum 120 over the ocean (30 years) and 72 over land (18 years) are used for the temporal

correlation analysis.

    As the reference height near the surface, we tested both $p_{ref} = p_{1000}$ and $p_{ref} = p_{sfc}$. In both cases, the density of the atmosphere within the lower troposphere below 700 hPa was set to $\rho = 1$ [$kg\ m^{-3}$]. When $p_{ref} = p_{1000}$, all thermodynamic properties at $p_{1000}$ (e.g., $p$, $\theta$, and $q_v$) are obtained from 6-hourly ERA Interim data interpolated into the time and location of individual EECRA observations. The geometric height of $p_{1000}$ from the surface ($z_{1000}$) is computed using a hydrostatic

equation. If the air at $p_{1000}$ is saturated, $z_{LCL}$ is set to $z_{1000}$. When $p_{ref} = p_{sfc}$, a set of thermodynamic properties at $p_{sfc}$ (e.g., $\theta$, $q_v$) is obtained from two different data sources: one is from the 6-hourly ERA Interim data interpolated into the time and location of individual EECRA observations similar to the case of $p_{ref} = p_{1000}$ and the other is from EECRA surface observations. When the surface air is saturated, we set $z_{LCL} = z_{sfc}$. Three separate analyses were performed using individual datasets (e.g., $p_{ref} = p_{1000}$, $p_{ref} = p_{sfc}$ with ERA Interim surface data, and $p_{ref} = p_{sfc}$ with EECRA surface observations).

In the next section, we will show the results based on $p_{ref} = p_{sfc}$ with EECRA surface observations of $p_{sfc}$, $\theta_{sfc}$, and $q_{v,sfc}$.





The analyses using the other two choices produced similar results. It is not shown here, but the same analysis using the NCEP/NCAR reanalysis product (Kalnay et al., 1996) instead of the ERA Interim data also produced similar results.

A fundamental assumption of our decoupling hypothesis is that thermodynamic scalars at the inversion base ($\theta_{inv}^-$) are bounded by the ML ($\theta_{ML}$) and inversion top ($\theta_{inv}^+$) properties. That is, the decoupling parameter $\alpha$ is between 0 and 1 [see Eqn.(3)]. However, the inversion height, $z_{inv}$ estimated from Eqn.(4) can have any numerical values, such that $\alpha$ parameterized by $z_{inv}$ (i.e., $\alpha = (z_{inv} - z_{ML})/\Delta z_s$) can be smaller than 0 or larger than 1. To be consistent with our decoupling hypothesis, we reset $z_{inv} = z_{ML}$ whenever $z_{inv}$ estimated from Eqn.(4) is smaller than $z_{ML}$ (i.e., we wrap all $\alpha < 0$ to $\alpha = 0$), and similarly, we reset $z_{inv} = \Delta z_s + z_{ML}$ whenever the estimated $z_{inv}$ is larger than $\Delta z_s + z_{ML}$ (i.e., we wrap all $\alpha > 1$ to $\alpha = 1$). As shown in Fig 2, the cases of $\alpha = 0$ after wrapping frequently occur in stably stratified regimes over the Arctic and northern continents in winter, desserts during the night, the northwestern Pacific and S.H. circumpolar regions during boreal summer when warm air is advected into cold SST, and along the west coast of major continents where cold SST exists due to coastal upwelling. On the other hand, the cases of $\alpha = 1$ after wrapping occur in unstable regimes over the tropical SST warm pools in JJA and along the midlatitude storm tracks during winter. In the next section, we will first present the results based on the data of $0 < \alpha < 1$ and then all data of $0 \leq \alpha \leq 1$ (e.g., the sum of $0 < \alpha < 1$, $\alpha = 0$, and $\alpha = 1$).

## 3 Results

### 3.1 Spatial-Seasonal Correlation

Figures 3 and 4 show the seasonal climatology of LCA (solid lines; also see Fig 7 for the color maps of LCA in the $2.5^o$latitude x $5^o$longitude grid box) and twelve proxies for LCA defined in the previous section during JJA and DJF, respectively. The spatial-seasonal correlation coefficients between these variables are summarized in Table 1. The scatter plots between the twelve proxies and LCA over the ocean and land are shown in Figures 5 and 6, respectively.

Consistent with KH93 and WB06, both LTS and EIS are strong in subtropical marine stratocumulus decks west of major continents but weak in the tropical deep convection regime in which LCA is small. Although the overall pattern of EIS looks similar to that of LTS, several notable differences exist between them. For example, in the vicinity of polar and high-latitude regions in both hemispheres where LCA is large, EIS is strong but LTS is weak, which seems to contribute to a low spatial-seasonal correlation between LTS and EIS over the globe (see Table 1). According to Eq.(2), EIS here should be weaker than in other regions, because $z_{LCL}$ is low in these cold regions (see Figs 3g and 4g). However, probably due to the decreases in $\Gamma_{700}^m$ and $z_{700}$ in the cold region, EIS becomes strong here, resulting in a better spatial-seasonal correlation between EIS and LCA ($r = 0.62$) than between LTS and LCA ($r = -0.05$) over the ocean. Table 1 shows that the spatial-seasonal correlation between LTS and LCA is very weak over the ocean and land. This is somewhat surprising, because KH93 reported that LTS is significantly positively correlated with LCA. It should be noted that KH93's finding was based on the analysis over the marine stratocumulus deck, while our result is based on global analysis. Because of this potential sensitivity of the relationship between LTS and LCA to the analysis domain, we need to be careful when using LTS as a global proxy for LCA in climate sensitivity studies.



The spatial pattern of the inversion strength (IS, Figs 3d and 4d), which is about 1-3 K higher than EIS, is roughly similar to that of EIS. However, in contrast to EIS but similar to LTS, IS tends to be small in high-latitude regions, which results in a weaker spatial-seasonal correlation between IS and LCA over the ocean ($r = 0.39$) than between EIS and LCA. Table 1 shows that the decoupling strength (DS, Figs 3e and 4e) is almost perfectly correlated with EIS over the globe ($r = -0.98$),
with a correlation with LCA very similar to that of EIS. Although named estimated inversion strength, EIS has a weaker correlation with inversion strength than anti-correlation with decoupling strength. In other words, a strong EIS indicates that the air at the inversion base is coupled well with the surface-based ML air. The decoupling parameter ($\alpha$, Figs 3f and 4f) also has a near-perfect correlation with EIS ($r = -0.98$), supporting our interpretation of EIS. When $\Gamma_{700}^m \approx \Gamma_{DL}^m$, it becomes $\alpha = DS/(\Gamma_{DL}^m \cdot \Delta z_s) \approx 1 - EIS/(\Gamma_{DL}^m \cdot \Delta z_s)$. As will be shown later, stronger correlation among EIS-DS-$\alpha$ than EIS-IS
also exists in the combined spatial-seasonal-interannual correlation statistics (Table 3) and in the analysis using all of the observation data (Tables 4 and 6).

The inversion height ($z_{inv} = \alpha \cdot \Delta z_s + z_{LCL}$, Figs 3h and 4h), which is strongly correlated with EIS, DS, and $\alpha$, has a high spatial-seasonal correlation with LCA ($r = -0.69$ over the ocean and $r = -0.77$ over land), even better than EIS ($r = 0.62$ over the ocean and $r = 0.07$ over land). The superiority of $z_{inv}$ over EIS as a proxy for LCA is more pronounced over land.
Why does $z_{inv}$ characterize LCA better than EIS over land ? If $\Gamma_{700}^m \approx \Gamma_{DL}^m$, $z_{inv} = \Delta z_s - EIS/\Gamma_{DL}^m + z_{LCL}$ such that the different correlation characteristics of $z_{inv}$ and EIS with LCA are likely associated with $z_{LCL}$. In the case that surface air is saturated and very low-level clouds are formed (e.g., fog with a low $z_{LCL}$), small $z_{LCL}$ causes EIS to decrease [EIS $\approx$ LTS $-\Gamma_{DL}^m \cdot (z_{700} - z_{LCL})$ from Eq.(2)]. This contributes to the weaker, negative EIS-LCA correlation seen over land compared to the stronger, positive correlation observed in the marine stratocumulus deck. Because $z_{inv}$ is defined as the addition of
$z_{LCL}$ to EIS, the undesirable negative impact of $z_{LCL}$ to the correlation with LCA is removed from EIS compared to $z_{inv}$. Over the western continental U.S. during summer, both $z_{LCL}$ and $z_{inv}$ are maximum, where LCA is at its minimum. In this region, similar to the ocean case, $z_{inv}$ is negatively correlated with LCA; however, opposite to the ocean case, EIS is not positively correlated with LCA. Consequently, EIS has a weaker global spatial-seasonal correlation with LCA ($r = 0.22$) than $z_{inv}$ ($r = -0.67$). It is very interesting to note that a simple proxy, $z_{LCL}$, shows a stronger spatial-seasonal (and also
spatial-seasonal-interannual) correlation with LCA than any of LTS, EIS, and $z_{inv}$ over both the ocean and land.

According to our conceptual framework, low-level clouds likely form just below $z_{inv}$ (see Fig 1). If that is the case, it is likely that the relative humidity at the base of $z_{inv}$ ($RH_{inv}^-$) is a good proxy for LCA. Over both the ocean and land, $RH_{inv}^-$ shows a stronger correlation with LCA than LTS and EIS but a weaker correlation than $z_{inv}$ and $z_{LCL}$. We speculate that this relatively poor performance of $RH_{inv}^-$ compared to $z_{inv}$ and $z_{LCL}$ is due in part to the poor estimation of $q_{v,inv}^-$, rather
than indicating that $RH_{inv}^-$ is a poor proxy for LCA. Because $RH_{inv}^-$ is estimated using the roughly estimated $\theta_{inv}^-$ and $q_{v,inv}^-$ without performing any saturation adjustment, the absolute values of $RH_{inv}^-$ in our analysis can be unreasonably larger than 100% in some regions (Figs 3i and 4i).

The spatial patterns of two LCS parameters, $\beta_1 = (z_{inv} + z_{LCL})/\Delta z_s$ and $\beta_2 = \sqrt{z_{inv} \cdot z_{LCL}}/\Delta z_s$, are roughly similar to those of $z_{inv}$ and $z_{LCL}$. However, both LCS parameters have a better spatial-seasonal correlation with LCA than $z_{inv}$ and
$z_{LCL}$, with $\beta_2$ showing a slightly better performance than $\beta_1$ (see Table 1 and Figs 5, 6). The estimated low-level cloud





fraction, ELF=$f(1-\beta_2)$ shows a very similar correlation as $\beta_2$ because the freezedry factor, $f$ is approximately 1 in the regime of $0 < \alpha < 1$ (see Fig 2). Among all twelve proxies, $\beta_2$/ELF has the highest spatial-seasonal correlation with LCA over both the ocean and land. ECTEI is equivalent if slightly better than EIS for the ocean, equivalent over land, and between LTS and EIS over the globe.

Figure 8 shows the scatter plots between the seasonal LCA and twelve proxies in several regions shown in Fig 7. LTS over land shows a stronger interregional-seasonal correlation with LCA than that over the ocean; however, EIS (and ECTEI, DS, $\alpha$) over the ocean has a stronger correlation than that over land. Over both the ocean and land, $z_{inv}$ and $z_{LCL}$ are strongly correlated with LCA. $RH^-_{inv}$ is strongly correlated with LCA over each of the ocean and land; however, the correlation over the globe is weak. Each of $\beta_1$ and $\beta_2$/ELF explains about 80% of interregional-seasonal variations of the seasonal LCA, which is a

much larger percentage than those explained by LTS (25%), EIS (25%), and ECTEI (7%). Except LTS, the difference between the regression slopes for the ocean and land in each proxy shown in Fig 8 is generally larger than the seasonal differences between the regression slopes shown in Figs 5 and 6, indicating a need to incorporate additional factors characterizing the contrast between the ocean and land in the future.

## 3.2    Seasonal-Interannual Correlation

Figure 9 shows the global distribution of seasonal-interannual correlation coefficients between the seasonal LCA and twelve proxies. Over the North Pacific and subtropical marine stratocumulus deck west of major continents, LTS and LCA are positively correlated. However, a negative correlation also exists, for example, over Europe and the Northwest Atlantic. Similar to LTS, EIS and LCA are positively correlated over marine stratocumulus deck; but interestingly, the correlation signs over the Northwest Atlantic and Europe and several continents (e.g., Southeast Asia, Australia, South America, and Central Africa) are reversed compared with LTS. As mentioned before, $z_{LCL}$ mainly contributes to the different correlation characteristics be-

tween EIS and LTS with LCA in these regions [see Eq.(2)]. Although LTS and EIS are good proxies for LCA over the marine stratocumulus deck, they have a limitation as a global proxy due to the spatial changes of temporal correlation signs. ECTEI shows a very similar correlation characteristics as EIS. With an opposite sign, the overall correlation patterns of DS and $\alpha$ are quite similar to that of EIS. Compared with the other proxies, LCA tends to be more homogeneously correlated with $z_{LCL}$,

$z_{inv}$, $RH^-_{inv}$, $\beta_1$, and $\beta_2$/ELF over the globe, without the spatial reversal of the correlation sign. Figure 10 shows a similar temporal correlation analysis as that shown in Fig 9 but only for interannual variations without the seasonal cycle. Overall, the spatial pattern of interannual correlation is similar to that of seasonal-interannual correlation; however, the magnitude is reduced, particularly, over the marine stratocumulus deck. Both in terms of seasonal-interannual and interannual correlations, $\beta_2$/ELF perform better than LTS and EIS over the entire globe without the spatial changes of temporal correlation signs; thus,

they are well suited as global proxies for LCA. $\beta_1$ performs nearly as well if not equivalently to $\beta_2$/ELF.

Table 2 summarizes the temporal correlation coefficients between LCA and twelve proxies for various regions shown in Fig 7. The first six regions in our analysis are the subtropical marine stratocumulus decks. Although less well-known than the other regions, stratocumulus and fog also exist over the Arabian Sea during summer and thus is included in our analysis (see Park and Leovy (2004) and Schubert et al. (1979) and also Fig 7a). The next three regions are the midlatitude marine stratocumulus decks



which have large LCA and undergo frequent passages of synoptic storms (North Pacific, North Atlantic, S.H. circumpolar). The last seven regions are over the continents and have small LCA. China is a unique continental stratocumulus deck with high LCA. Over the marine stratocumulus decks, except in the S.H. circumpolar region, LTS has a significant positive seasonal-interannual correlation with LCA. In all regions except the Arabian Sea, both EIS and ECTEI show strong correlations similar

to that of LTS. Although mainly designed as a proxy for the marine stratocumulus fraction, LTS is also positively correlated with some continental LCA in a statistically significant way. On the other hand, as is also shown in Fig 9, EIS and ECTEI are strongly negatively correlated with the continental LCA over India, South America, and the Southwest Sahara. Similar to the spatial-seasonal correlations, the temporal correlation characteristics of DS and $\alpha$ with LCA are very similar to that of EIS. Over the stratocumulus decks, $z_{inv}$ is a proxy that is as good as or better than LTS and EIS. Notably, $z_{LCL}$ shows a very good

performance over land (and also in the midlatitude marine stratocumulus decks), indicating that surface moisture (regardless of whether it is locally-originated or advected) substantially contributes to the temporal variations of the continental LCA. In most areas, $RH_{inv}^{-}$ has a significant positive seasonal-interannual correlation with LCA. In general, the interannual correlation tends to be weaker than the seasonal-interannual correlation. In the subtropical marine stratocumulus decks, except the Namibian and Australian, the seasonal cycle dominantly contributes to the temporal correlation between various proxies and LCA. Regardless

of whether the seasonal cycle is included or not, two LCS parameters $(\beta_1, \beta_2)$ and ELF have better temporal correlations with LCA than any other proxies over almost all regions, including the marine stratocumulus decks.

Table 3 summarizes the combined spatial-seasonal-interannual correlations between the seasonal LCA and twelve proxies. Over the ocean, EIS and ECTEI show better correlations with LCA than LTS. EIS correlates better with DS and $\alpha$ than with IS. As a global proxy for LCA, both $z_{LCL}$ and $z_{inv}$ perform better than LTS, EIS and ECTEI because of their better performance

over land. A very simple proxy, $z_{LCL}$ performs better than $z_{inv}$ and explains almost 50% of the spatiotemporal variations of LCA over the globe. Among all twelve proxies, $\beta_2$/ELF explain the largest fraction of spatial and temporal variations of the seasonal LCA (more than 56 % over the entire globe), much more than the ones explained by LTS (3%) and EIS (4%). Over the ocean, $\beta_2$ performs slightly better than $\beta_1$ with a similar performance over land.

### 3.3  Extended Analysis using All Data

Until now, we have presented results based on the analysis of the data satisfying $0 < \alpha < 1$ (i.e., $z_{LCL} < z_{inv} < \Delta z_s + z_{LCL}$) when our decoupling hypothesis can be applied without any conceptual ambiguity. However, for a fair comparison with the previous proxies of LTS, EIS, and ECTEI, which can be defined in any cases, it is necessary to examine the performance of various proxies in general situations. This section provides an extended analysis using all observation data of $0 \le \alpha \le 1$ (i.e., the sum of $0 < \alpha < 1$, $\alpha = 0$, and $\alpha = 1$). Figure 2 shows the frequency of occurrence of $\alpha = 0$ and $\alpha = 1$ among all observations

during JJA and DJF. The cases of $\alpha = 0$ frequently occur when the lower-troposphere is highly stable (i.e., $z_{inv} \le z_{LCL}$) in the vicinity of the Arctic area north of $60^o$N, the northern continents and desserts during boreal winter, the northwestern Pacific and S.H. circumpolar regions during boreal summer, and along the west coast of major continents. On the other hand, the cases of $\alpha = 1$ occur when the troposphere is highly unstable (i.e., $z_{inv} \ge \Delta z_s + z_{LCL}$) in the tropical warm pool regions during JJA and the midlatitude storm tracks during boreal winter. The differences between the results presented in this section and





previous sections are due to the inclusion of the data obtained from these highly stable ($\alpha = 0$) and unstable regimes ($\alpha = 1$), mostly from stable regimes. This comparison allows us to obtain insights into the impact of extreme vertical stratification in the lower-troposphere on the relationship between various proxies and LCA.

Figure 11 shows the seasonal climatology of LCA (solid lines) and six key proxies (LTS, EIS, ECTEI, $\beta_1$, $\beta_2$, and ELF)
during JJA and DJF, respectively, for all observation data. The overall pattern of the climatological proxies is similar to those with $0 < \alpha < 1$ (Figs 3, 4), but the magnitude in the high latitude regions is amplified due mainly to the contribution of the data in the very stable regimes. In particular, over the northern continents during DJF, the values of LTS/EIS/ECTEI ($\beta_1/\beta_2$) are substantially higher (lower) than those of $0 < \alpha < 1$. The spatial-seasonal correlation between LTS and EIS ($r = 0.85$ in Table 4) is now much stronger than the previous case of $0 < \alpha < 1$ ($r = -0.08$ in Table 1), but in contrast to EIS, LTS still tends to
decrease with latitude in the S.H. high-latitude regions.

Figure 12 shows the scatter plots between LCA and six key proxies over the ocean, land, globe, and several regions defined in Fig 7. This is the reproduction of Figs 5, 6, and 8 using all observation data. Except ELF, the correlations between various proxies and LCA are degraded by the inclusion of the data from the very stable and unstable regimes. In particular, EIS and ECTEI over the ocean and two LCS parameters ($\beta_1$ and $\beta_2$) show substantial degradation of the correlation with LCA, due
mainly to the enhanced scatters induced by the data in the very stable regimes where LCA is small (the light-colored data points of $0 \leq \alpha < 0.01$ in Fig 12). Very surprisingly, however, ELF continues to maintain a strong correlation relationship with LCA over both the ocean and land, explaining 70% of spatial-seasonal variations of LCA over the entire globe, which is a much larger percentage than those explained by $\beta_1$ (19%), $\beta_2$ (30%), LTS (5%), EIS (6%), and ECTEI (4%). In comparison with the scatter plots of individual $5^o$lat x $10^o$lon grid point data, the scatter plots of the selected regions (the last column of
Fig 12) show weaker degradation of correlations because the selected regions are relatively free from the occurrence of $\alpha = 0$ and $\alpha = 1$ (compare Fig 7 with Fig 2a, b).

Table 4 is the reproduction of Table 1 using all the data of $0 < \alpha < 1$, $\alpha = 0$, and $\alpha = 1$. Except the IS over the ocean and ELF, the correlations between the proxies and LCA are generally degraded in comparison to the previous case of $0 < \alpha < 1$. The spatial-seasonal correlations between LTS (and EIS, ECTEI) and LCA are very weak and even tend to be negative. Similar
to the previous case, EIS is more strongly correlated with DS and $\alpha$ than with IS. Although it shows a weaker correlation than that in Table 1, $z_{LCL}$ still remains as a good proxy for LCA. Due mainly to the rapid decrease in the correlation between $z_{inv}$ and LCA, the correlations between two LCS parameters and LCA are substantially weaker than the previous case of $0 < \alpha < 1$. However, ELF successfully maintains a very strong spatial-seasonal correlation with LCA over both the ocean and land.

Figure 13 shows the global distribution of seasonal-interannual (upper) and interannual (lower) correlation coefficients be-
tween the seasonal LCA and six key proxies. The overall correlation patterns over the ocean are similar to the previous cases of $0 < \alpha < 1$ (see Figs 9 and 10). However, over Asia, LTS/EIS/ECTEI show very strong negative seasonal-interannual correlations, resulting in the opposite correlations between the ocean and land. Over Asia, with similar contributions from $z_{inv}$ and $z_{LCL}$ (not shown but more strongly from $z_{inv}$ for seasonal-interannual but more strongly from $z_{LCL}$ for interannual correlations), both $\beta_1$ and $\beta_2$ show undesirable positive seasonal-interannual correlations with LCA. This indicates that LCS, similar
to EIS/LTS/ECTEI, is not appropriate as a global proxy for diagnosing seasonal-interannual variations of LCA. However, LCS





is still a useful global proxy for diagnosing interannual variations of LCA. Among all the proxies examined, ELF remains the best proxy in terms of diagnosing both the seasonal-interannual and interannual variations of LCA over the globe, including the marine stratocumulus deck.

Table 5 is the reproduction of Table 2 using all data of $0 \leq \alpha \leq 1$. Similar to Table 2, LTS remains a good proxy for diagnos-
ing the seasonal-interannual variations of LCA in the subtropical marine stratocumulus deck. However, its performance over several land areas (e.g., South America, Australia, Southwest Sahara, India, and China) is degraded, but its performance over Europe is improved. Both EIS and ECTEI show similar correlation characteristics as those in Table 2, but the undesirable negative correlation in the western US is changed to a desirable strong positive correlation. LTS outperforms EIS and ECTEI over the Arabian marine stratocumulus deck and India. LTS, EIS, and ECTEI all show undesirable significant negative correlations with LCA in the S.H. circumpolar region. In the stratocumulus deck (and also over most land areas), $z_{inv}$ remains as a proxy that is as good as or better than LTS and EIS. Over the land and midlatitude stratocumulus deck, including the Arabian region, $z_{LCL}$ tends to work better than $z_{inv}$. Similar to Table 2, ELF and two LCS parameters perform better than LTS and EIS in most regions, including the marine stratocumulus deck. Overall, among the twelve proxies, ELF shows the best performance in diagnosing the seasonal-interannual variations of the seasonal LCA in a statistically significant way. The only exception is continental Australia in which both $z_{LCL}$ and $z_{inv}$ are weakly correlated with LCA. In addition to the combined seasonal-interannual variations, ELF also diagnoses the interannual variations of LCA well. However, similar to LTS and EIS, ELF still does not show good performance in diagnosing the interannual variations of LCA in several subtropical marine stratocumulus decks in a statistically significant way.

Table 6 summarizes the combined spatial-seasonal-interannual correlations between the seasonal LCA and twelve proxies in general cases with all observation data. LTS is poorly correlated with LCA. Over the ocean, EIS and ECTEI show slightly better performance than LTS, but the improvement seems to be marginal. EIS correlates much better with DS and $\alpha$ than with IS. Among all twelve proxies, ELF shows the best performance in diagnosing the spatiotemporal variations of LCA, explaining almost 60 % of the spatial-seasonal-interannual variance of the seasonal LCA over the entire globe (50 and 62 % over the ocean and land, respectively), which is a much larger percentage than those explained by LTS (2%), EIS (4%), and ECTEI (2%).

## 4   Summary and Conclusion

Based on the decoupling parameterization of the cloud-topped PBL suggested by PLR04, a simple heuristic equation is derived to compute the inversion height, $z_{inv}$. As an attempt to find a simple heuristic proxy for diagnosing the spatiotemporal variations of LCA over the globe, we defined two low-level cloud suppression parameters (LCS), $\beta_1 = (z_{inv} + z_{LCL})/\Delta z_s$ and $\beta_2 = \sqrt{z_{inv} \cdot z_{LCL}}/\Delta z_s$ by combining $z_{inv}$ with the lifting condensation level of near-surface air, $z_{LCL}$, and normalizing with a constant scale height, $\Delta z_s = 2750$ [m]. To better diagnose LCA in extremely cold and dry atmospheric conditions, we also defined an estimated low-level cloud fraction, ELF$\equiv f(1 - \beta_2)$ with a freezedry factor, $f = max[0.15, min(1, q_{v,ML}/0.003)]$, where $q_{v,ML}$ is the water vapor specific humidity in the surfaced-based mixed layer that is assumed to be topped by $z_{LCL}$, for simplicity. If $z_{ML} \approx z_{LCL}$ where $z_{ML}$ is the height of surface-based mixed layer, then $\beta_1 \approx (z_{LCL}/\Delta z_s) \cdot [2 + (z_{inv} - $



$z_{ML})/z_{ML}]$ and $\beta_2 \approx (z_{LCL}/\Delta z_s) \cdot \sqrt{1 + (z_{inv} - z_{ML})/z_{ML}}$, where the first factor in both formulae represents the degree of subsaturation of near-surface air, and the second factor represents the decoupling strength. ELF predicts that LCA increases as the inversion base air is thermodynamically coupled with the moist near-surface air containing enough amount of water vapor. Considering that low-level clouds usually form at the inversion base, the conceptual cloud formation processes embedded in
ELF are consistent with what is expected to happen in nature.

Using the near surface $\theta$ and $q_v$ obtained from individual EECRA surface observations and the 6-hourly ERA Interim profile of $\theta$, we computed the IS (inversion strength), DS (decoupling strength), $\alpha$ (decoupling parameter), $z_{LCL}$, $z_{inv}$, $RH^-_{inv}$, $\beta_1$, $\beta_2$, and ELF for January 1979-December 2008 over the ocean and January 1979-December 1996 over land, respectively, which were then averaged into $5^o$latitude x $10^o$longitude seasonal grid data. Spatial and temporal correlations between these proxies
and surface-observed seasonal LCA were computed over the globe and compared with those of LTS, EIS, and ECTEI, all of which have been widely used as proxies for LCA in previous studies. To obtain insights into the impact of extreme vertical stratification in the lower-troposphere (i.e., very stable or unstable regimes) on the correlation relationship between LCA and various proxies, we first analyzed the results only using the data satisfying $0 < \alpha < 1$ (i.e., $z_{LCL} < z_{inv} < \Delta z_s + z_{LCL}$) and then the analysis was extended to include all of the observation data. Here, we provide a summary of the results for $0 < \alpha < 1$
and then for all data.

In contrast to previous studies, the spatial-seasonal correlation between LTS and LCA is very weak, reflecting the sensitivity of LTS-LCA relationship to the analysis domain. However, EIS and ECTEI show a significant positive correlation with LCA over the ocean (Table 1 and Figs 5, 6). It is shown that EIS is more strongly anti-correlated with $\alpha$ and DS than with IS, implying that EIS may measure the magnitude of decoupling strength more strongly than the inversion strength. As a proxy
for diagnosing spatial-seasonal variations of LCA, $z_{LCL}$ performs better than $z_{inv}$ and $RH^-_{inv}$, which work better than LTS. The LCS parameter, $\beta_2$, performs slightly better than $\beta_1$ which is better than $z_{LCL}$. Among all twelve proxies, ELF shows the best performance. A similar interregional-seasonal correlation analysis over several selected regions (Fig 8) also showed that $\beta_1$ and $\beta_2$/ELF are the best proxies for LCA, explaining about 80% of interregional-seasonal variance of the seasonal LCA, much higher than the ones explained by LTS and EIS (25%).

In addition to the spatial-seasonal correlation, we also examined the seasonal-interannual and interannual correlations between the proxies and LCA (Table 2 and Figs 9, 10). Consistent with previous studies, both LTS and EIS are good proxies characterizing the seasonal-interannual variation of LCA over subtropical marine stratocumulus decks. Although mainly designed as a proxy for marine stratocumulus, LTS is also positively correlated with some continental LCA. Due to the spatial changes of temporal correlation signs, however, the use of LTS and EIS as global proxies for LCA is limited. In contrast to
LTS and EIS, the proxies of $z_{inv}$, $z_{LCL}$, $RH^-_{inv}$, $\beta_1$, and $\beta_2$/ELF tend to be more homogeneously correlated with LCA all over the globe, without the reversal of correlation signs (Fig 9). Similar to the spatial-seasonal correlations, the temporal correlation characteristics of DS and $\alpha$ with LCA are very similar to that of EIS. The interannual correlation between the proxies and LCA tends to be weaker than the seasonal-interannual correlation, and all proxies over the subtropical marine stratocumulus decks except the Namibian have weak interannual correlations smaller than 0.5 with LCA (Table 2 and Fig 10). Regardless
of whether the seasonal cycle is included or not, $\beta_2$/ELF show the best temporal correlation with LCA in almost all regions.





The combined spatial-seasonal-interannual correlation analysis reveals that EIS and ECTEI over the ocean are significantly correlated with LCA but LTS is poorly correlated with LCA (Table 3). Among all twelve proxies, $\beta_2$/ELF explain the largest fraction of spatial and temporal variations of the seasonal LCA over the globe.

For a fair comparison with the previous proxies of LTS, EIS, and ECTEI that can be defined in any situations, we repeated our analysis by using all observation data, including the ones in very stable and unstable regimes (Tables 4-6 and Figs 11-13). In general, in comparison with the previous case of $0 < \alpha < 1$, the correlations between the proxies and LCA are degraded mainly due to the enhanced scatters induced by the data in the very stable regimes (Fig 12). However, ELF continues to maintain a very strong correlation with LCA. LTS, EIS and ECTEI remain good proxies for diagnosing the seasonal-interannual variations of LCA in the subtropical marine stratocumulus deck with better performance of LTS than EIS and ECTEI over the Arabian and Canarian regions (Table 5). However, they show undesirable negative correlations with LCA in the S.H. circumpolar region and several continental regions, particularly, Asia (Fig 13). Over the stratocumulus deck and most land areas except Asia, $z_{inv}$ remains a proxy that is as good as or better than LTS and EIS, while $z_{LCL}$ generally works better than $z_{inv}$ over land except the eastern U.S. EIS anti-correlates much better with DS and $\alpha$ than with IS. Among all twelve proxies, ELF shows the best performance in diagnosing the spatiotemporal variations of LCA (Table 6), explaining almost 60% of spatial-seasonal-interannual variances of the seasonal LCA over the entire globe (50 and 62% over the ocean and land, respectively), which is a much larger percentage than those explained by LTS (2%), EIS (4%), and ECTEI (2%). However, similar to LTS and EIS, ELF still has a weakness in diagnosing interannual variation of LCA in several subtropical marine stratocumulus decks in a statistically significant way (Table 5).

We have shown that ELF and two LCS parameters are superior to the previously proposed LTS, EIS, and ECTEI in diagnosing the spatial and temporal variations of the seasonal LCA over both the ocean and land, including the marine stratocumulus deck. However, there are a couple of aspects that need to be addressed in future research. First, mainly due to insufficient surface moisture, the physical processes controlling the formation and dissipation of LCA over land are likely to be different from those over the ocean [e.g., Ek and Mahrt (1994), Ek and Holtslag (2004), Zhang and Klein (2010), Zhang and Klein (2013), Gentine et al. (2013)]. In contrast to ocean areas where $z_{ML} \approx z_{LCL}$, $z_{ML}$ over desserts during the night is likely to be lower than $z_{LCL}$ due to the radiative stabilization of near-surface air. In addition, the mixed layer developed during the daytime may reside above the stable nocturnal surface layer; therefore, the idealized vertical structure depicted in Fig 1 may not be valid. At this stage, we are not aware of any comprehensive ways to handle these complicated cases over land within the degree of complexity suitable for a heuristic proxy or simple parameterization. One very simple way is to impose a certain upper limit on $z_{ML}$ [e.g., $z_{ML} = min(z_{LCL}, 1500\,m)$], but it would be more desirable to parameterize $z_{ML}$ as a function of the buoyancy flux and shear production within the PBL. Second, although named an estimated low-level cloud fraction due mainly to the existence of an upper bound of 1 (when near surface air is saturated with enough amount of water vapor), our ELF has systematic biases against LCA. For example, as seen in Figs 5l, 6l, 8l, and 12u-x, the dashed gray lines representing LCA=ELF are offset from the thick black regression lines, and the regression slope over the ocean is different from that over land. As a result, our ELF tends to overestimate LCA over land and to underestimate it over the marine stratocumulus deck (Figs 2g,h). This feature may be addressed by using different scale heights ($\Delta z_s$) over the ocean and land, respectively. Finally, a strong




correlation relationship between LCA and ELF (and two LCS parameters) identified in our study can be used to evaluate the realism of simulated LCA in GCMs, as was done by Park et al. (2014) using LTS. Because the freezedry factor is derived from the analysis of observation data, and the definitions of the observed and simulated LCA may differ, it might be better to use the LCS parameters ($\beta_1$ or $\beta_2$) instead of ELF to evaluate the simulated LCA, unless a GCM has a cloud fraction parameterization incorporating the freezedry factor. It was not shown here, but we checked that the observed significant correlations between LCS and LCA were also simulated by the Community Atmosphere Model version 5 (CAM5, Park et al. (2014)) and the Seoul National University Atmosphere Model version 0 with a Unified Convection Scheme [SAM0-UNICON, Park (2014a), Park (2014b), Park et al. (2017)], which will be reported in a separate paper with additional observational analysis by cloud types (e.g., cumulus, cumulonimbus, stratus, stratocumulus, and fog).

## 5 Implication

Our study implies that regardless of other properties, accurate prediction of inversion base height and lifting condensation level is a key factor necessary for successful simulation of global low-level clouds in weather prediction models and GCMs. Strong spatiotemporal correlation between ELF (or LTS) and LCA identified in our study can be used to evaluate the performance of GCMs, identify the source of inaccurate simulation of LCA, and better understand climate sensitivity.

*Author contributions.* Sungsu Park has led the overall research and Jihoon helped to conduct the analysis under the supervision of Sungsu Park.

*Competing interests.* The authors declare that they have no conflict of interest.

*Data availability.* The EERCA cloud data used in our study is available at https://rda.ucar.edu/datasets/ds292.2/

*Acknowledgements.* The first author expresses his deepest thanks to the late Prof. Conway B. Leovy at the University of Washington, the first author's Ph.D advisor, who developed the concept of PLR04's decoupling parameterization on which our derivation of ELF and LCS is based. This work was supported by the Creative-Pioneering Researchers Program of Seoul National University (SNU; 3345-20180017) and Research Resettlement Fund for new faculty of Seoul National University (SNU; 3345-20150014).





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




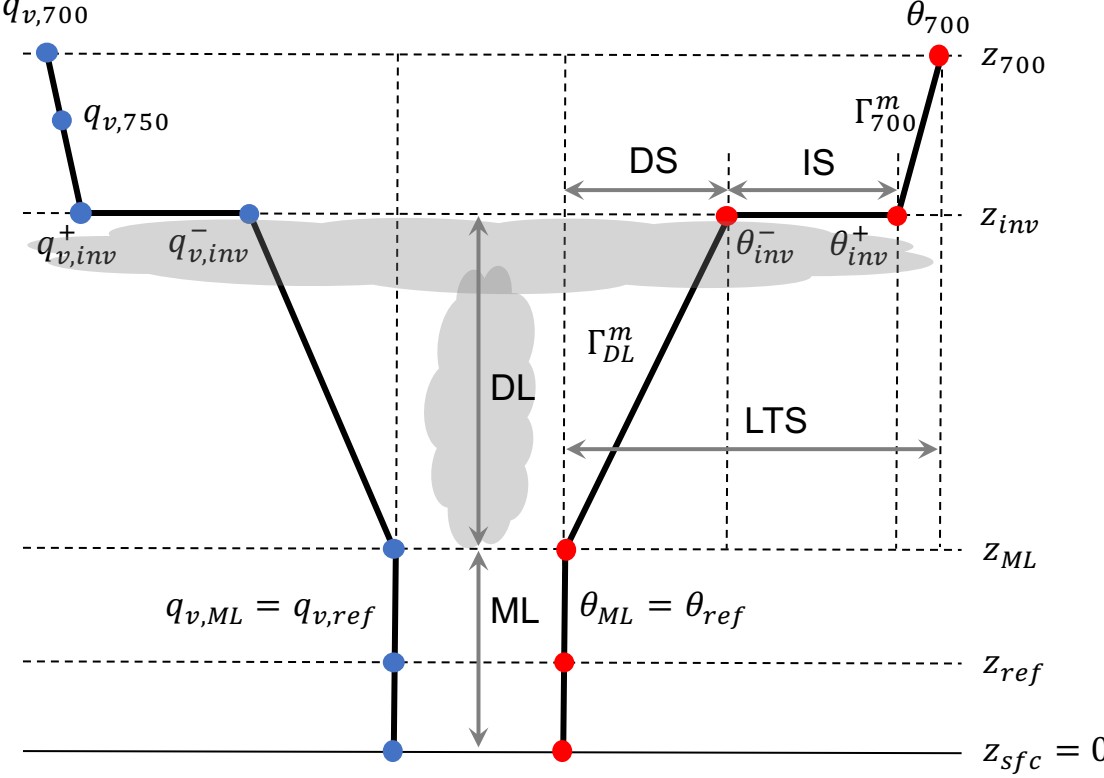

**Figure 1.** Schematic diagram illustrating an idealized structure of the decoupled planetary boundary layer (PBL), which consists of a surface-based mixed layer (ML) topped by $z_{ML}$ with the potential temperature, $\theta_{ML} = \theta_{ref}$ and water vapor specific humidity, $q_{v,ML} = q_{v,ref}$ given at the reference height, $z_{ref}$; a decoupled layer (DL) in which the vertical temperature gradient is assumed to follow a saturated moist adiabat at the ML top ($\Gamma_{DL}^m$); an inversion at $z_{inv}$ across which $\theta$ increases and $q_v$ decreases in a discontinuous way; and the free atmosphere between $z_{inv}$ and $z_{700}$ in which the vertical temperature gradient is assumed to follow a saturated moist adiabat at the 700 hPa level ($\Gamma_{700}^m$). Also shown are the inversion strength, IS $\equiv \theta_{inv}^+ - \theta_{inv}^-$; decoupling strength, DS $\equiv \theta_{inv}^- - \theta_{ML}$; and lower-tropospheric stability, LTS $\equiv \theta_{700} - \theta_{ML}$. The lapse rate of $q_v$ in the free atmosphere is given by the linear slope between $q_{v,700}$ and $q_{v,750}$. For simplicity, it is assumed that $\theta$ and $q_v$ share a common decoupling parameter, $\alpha \equiv (\theta_{inv}^- - \theta_{ML})/(\theta_{inv}^+ - \theta_{ML}) = (q_{v,inv}^- - q_{v,ML})/(q_{v,inv}^+ - q_{v,ML})$. Over the ocean, stratocumulus generally exists below the inversion base and shallow cumulus grows into the overlying stratocumulus in the decoupled layer. See the text for more details.





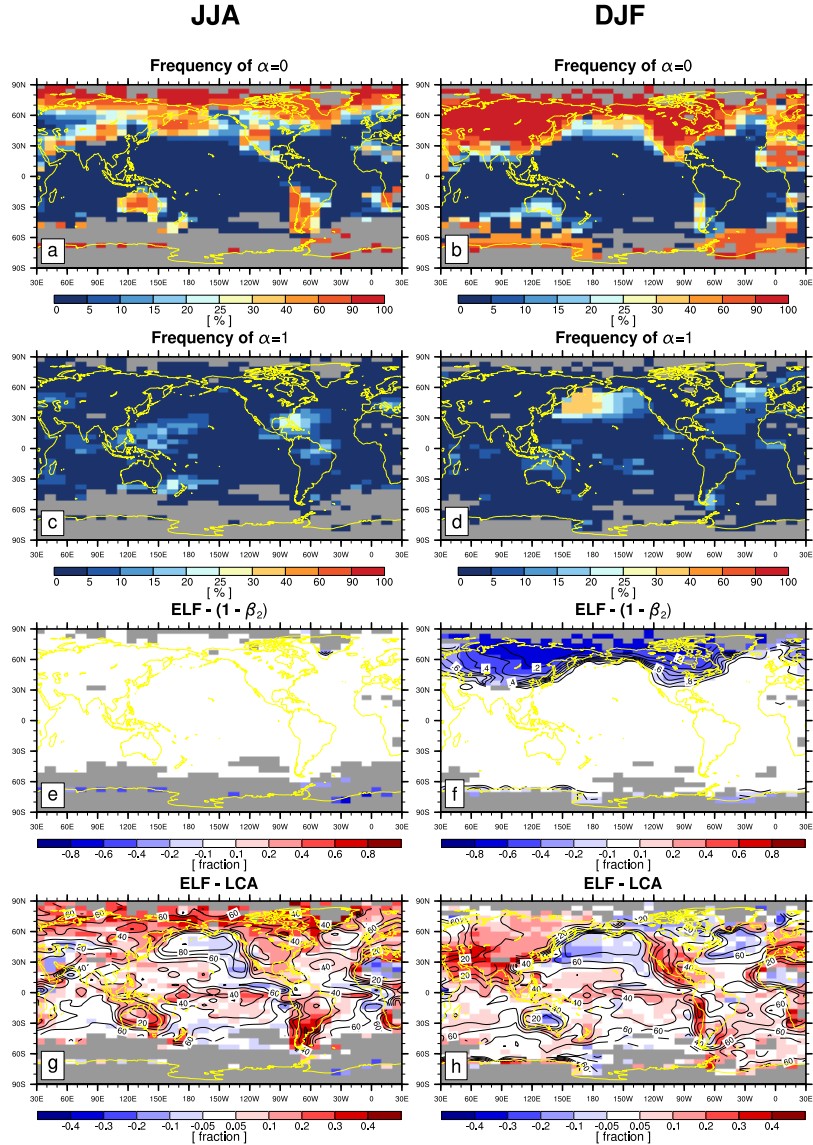

**Figure 2.** [(a),(b)] Frequency of occurrence of $\alpha = 0$ (i.e., $z_{inv} \leq z_{LCL}$) among all observations (the sum of $0 < \alpha < 1$, $\alpha = 0$, and $\alpha = 1$); [(c),(d)] frequency of occurrence of $\alpha = 1$ (i.e., $z_{inv} \geq \Delta z_s + z_{LCL}$) among all observations; [(e),(f)] the difference between ELF and $1 - \beta_2$ = ELF/$f$, where $f$ is the freezedry factor (see Eqns.(9),(10)); and [(g),(h)] the difference between ELF and LCA during (left column) JJA and (right column) DJF. In panels (e) and (f), the contour lines denotes the freezedry factor, $f$. In panels (g) and (h), the contour lines denotes the climatological LCA reported by surface observers. All observations are used for the panels (e)-(h). The grid boxes with a climatological observation number below 100 in each season are masked by grey color.





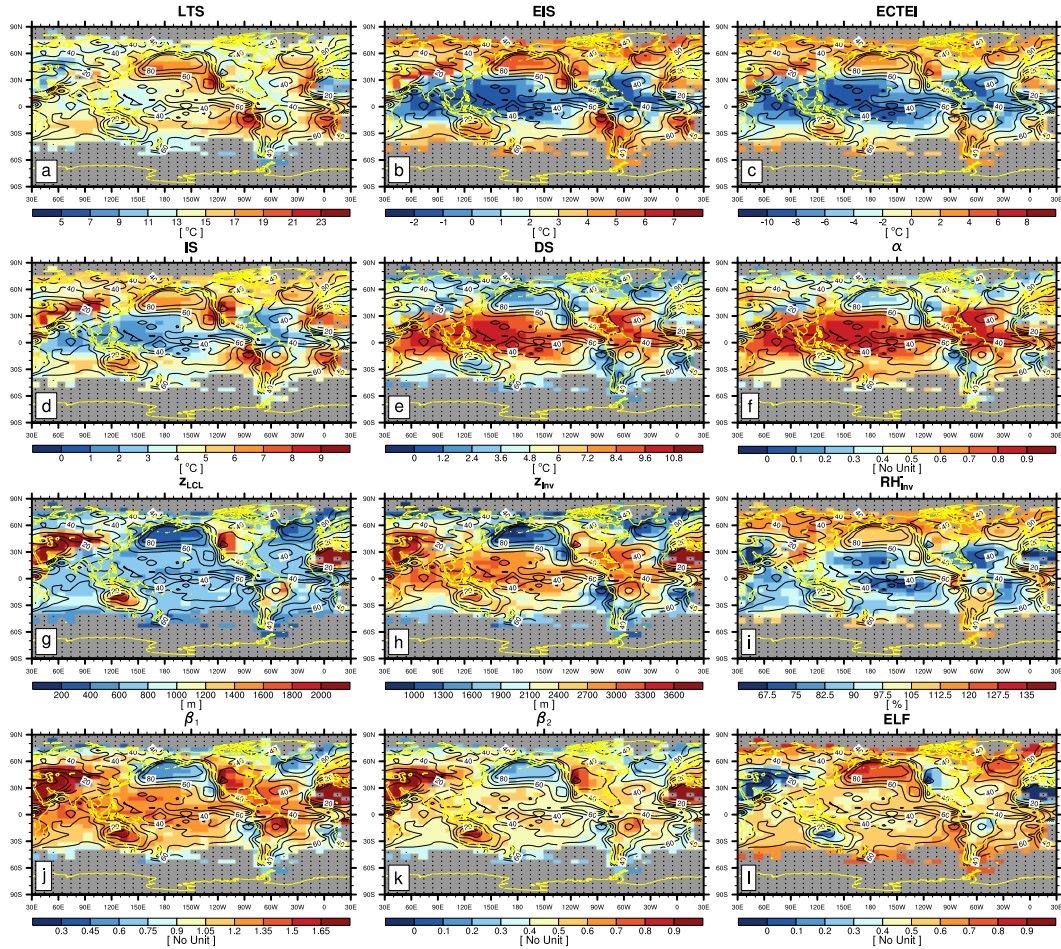

**Figure 3.** Global distribution of climatological (a) lower-tropospheric stability (LTS), (b) estimated inversion strength (EIS), (c) estimated cloud-top entrainment index (ECTEI), (d) inversion strength (IS), (e) decoupling strength (DS), (f) decoupling parameter ($\alpha$), (g) lifting condensation level of near-surface air ($z_{LCL}$), (h) inversion height ($z_{inv}$), (i) relative humidity at the inversion base ($RH_{inv}^{-}$), (j) the first low-level cloud suppression parameter ($\beta_1$), (k) the second low-level cloud suppression parameter ($\beta_2$), and (l) estimated low-level cloud fraction (ELF) in June-July-August (JJA). The contour line denotes the climatological LCA in JJA reported by surface observers. The grid boxes with a climatological observation number below 100 in JJA are masked by grey color and dots. Only the data satisfying $0 < \alpha < 1$ are used for this figure.





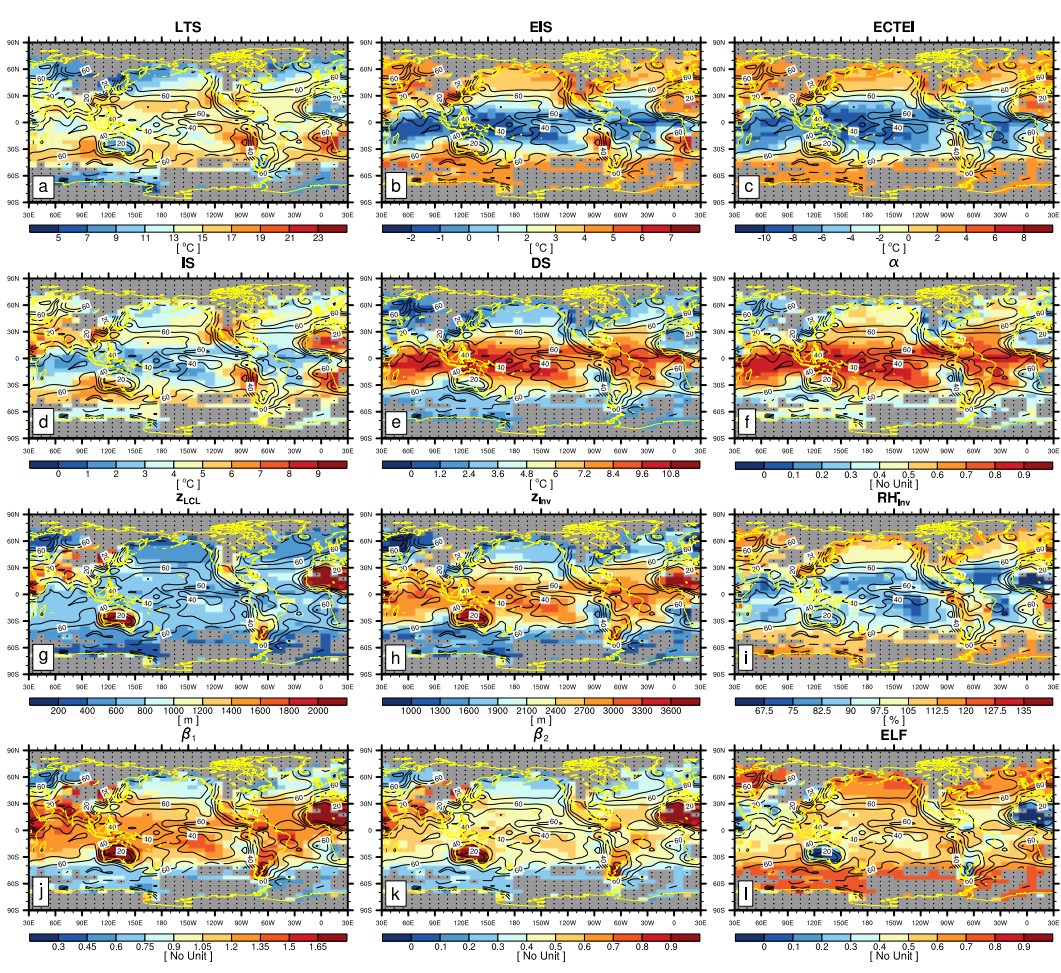

**Figure 4.** The same as Figure 3 but in December-January-February (DJF).





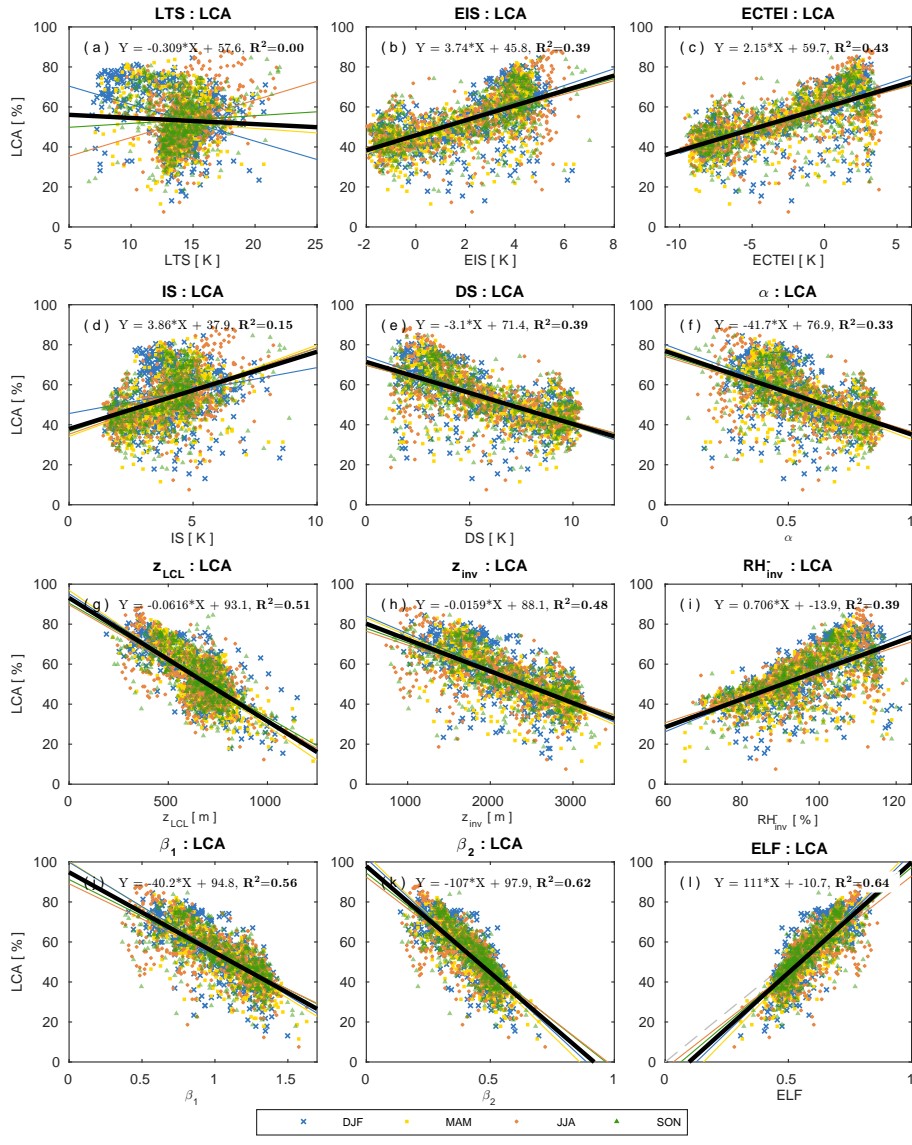

**Figure 5.** Scatter plots between LCA and various proxies over the ocean. Each dot reflects climatological seasonal data at each $5^o$ latitude x $10^o$ longitude grid box shown in Figs 3 and 4, and different colors denote different seasons. Also plotted are the least square fitting lines for each season (colored lines) and all seasons (thick black line) and regression equation for all seasons with the corresponding fraction of variance ($R^2$) explained by the all-seasons regression equation. The grey dashed line in the panel (l) denotes LCA = ELF. Only the data satisfying $0 < \alpha < 1$ are used for this figure.





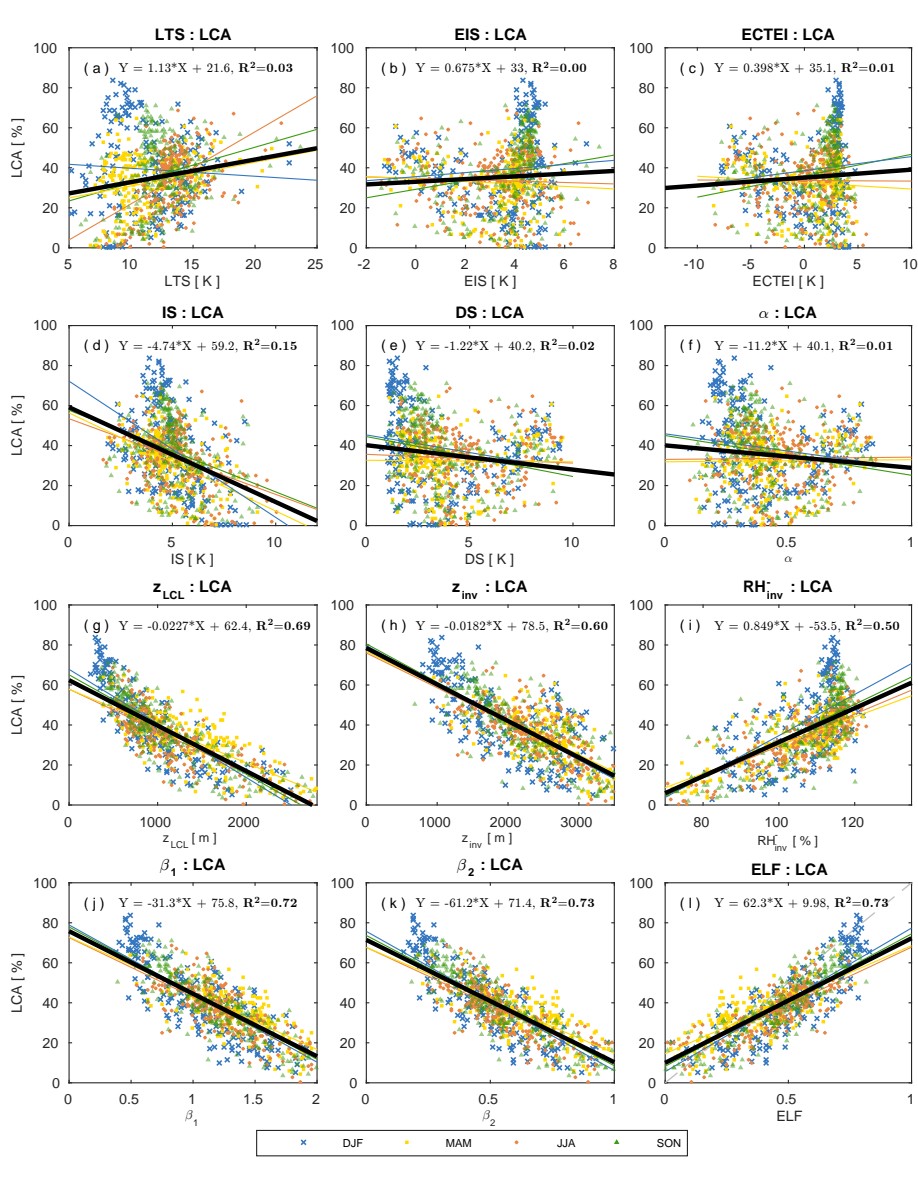

**Figure 6.** The same as Figure 5 but over land.




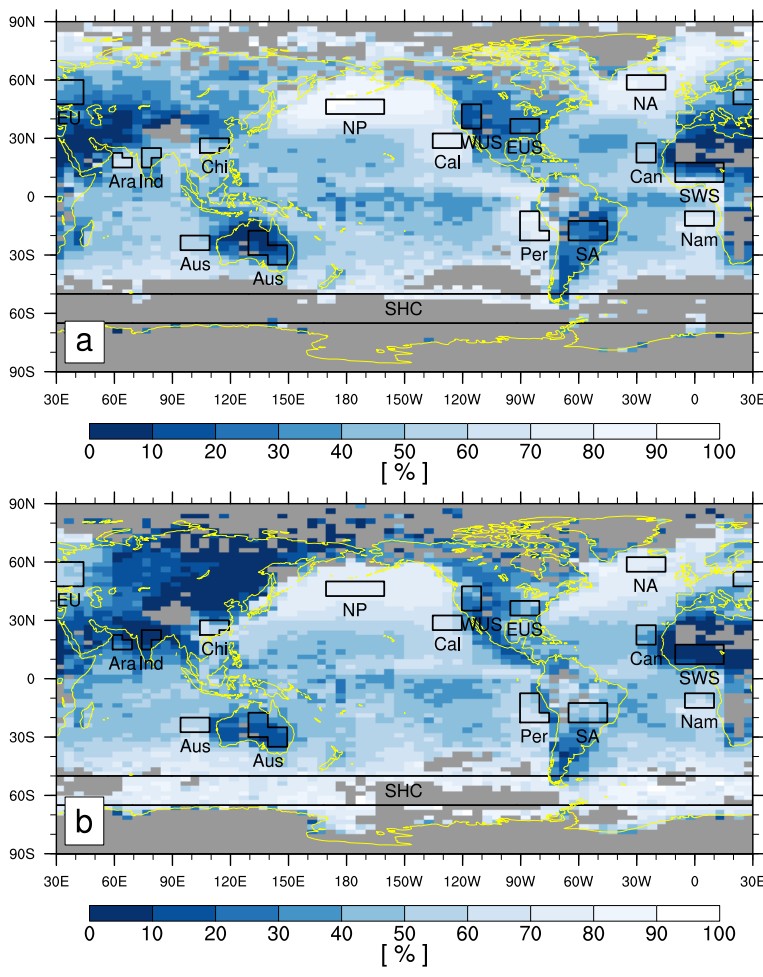

**Figure 7.** The seasonal climatology of LCA during (a) JJA and (b) DJF in the 2.5$^o$latitude x 5$^o$longitude grid box. The grid boxes with a climatological observation number below 25 in each season are masked by grey color. The domains selected for computing temporal (seasonal-interannual and interannual only) correlation coefficients in Tables 2 and 5 are also plotted: Peruvian [(7.5$^o$S-22.5$^o$S, 80$^o$W-90$^o$W) and (17.5$^o$S-22.5$^o$S, 75$^o$W-80$^o$W)], Namibian (7.5$^o$S-15$^o$S, 5$^o$W-10$^o$E), Californian (25$^o$N-32.5$^o$N, 120$^o$W-135$^o$W), Australian (20$^o$S-27.5$^o$S, 95$^o$E-110$^o$E), Canarian (17.5$^o$N-27.5$^o$N, 20$^o$W-30$^o$W), Arabian [(15$^o$N-20$^o$N, 60$^o$E-70$^o$E) and (20$^o$N-22.5$^o$N, 60$^o$E-65$^o$E)], North Pacific (42.5$^o$N-50$^o$N, 170$^o$E-200$^o$E), North Atlantic (55$^o$N-62.5$^o$N, 15$^o$W-35$^o$W), Southern-Hemispheric Circumpolar region (50$^o$S-65$^o$S, 0$^o$E-360$^o$E), China [(25$^o$N-30$^o$N, 105$^o$E-120$^o$E) and (22.5$^o$N-25$^o$N, 105$^o$E-115$^o$E)], India [(20$^o$N-25$^o$N, 75$^o$E-85$^o$E) and (15$^o$N-20$^o$N, 75$^o$E-80$^o$E)], Europe [(47.5$^o$N-60$^o$N, 30$^o$E-45$^o$E) and (47.5$^o$N-55$^o$N, 20$^o$E-30$^o$E)], Eastern US (32.5$^o$N-40$^o$N, 80$^o$W-95$^o$W), Western US (35$^o$N-47.5$^o$N, 110$^o$W-120$^o$W), South America (12.5$^o$S-22.5$^o$S, 45$^o$W-65$^o$W), Australia [(25$^o$S-35$^o$S, 140$^o$E-150$^o$E) and (17.5$^o$S-30$^o$S, 130$^o$E-140$^o$E)], and Southwest Sahara (7.5$^o$N-17.5$^o$N, 10$^o$W-15$^o$E).







**Figure 8.** Scatter plots between the seasonal LCA and various proxies over selected regions of Figure 7. Different colors denote different seasons. The ocean and land regions are denoted by dots and x marks, respectively, except for the China stratocumulus deck, which is denoted by the circled x marks. Also plotted are the least square fitting lines for ocean (O; dashed line), land (L; dotted line), and globe (G; thick solid line), respectively, with the fraction of variance ($R^2$) explained by the regression lines. Only the data satisfying $0 < \alpha < 1$ are used for this figure.




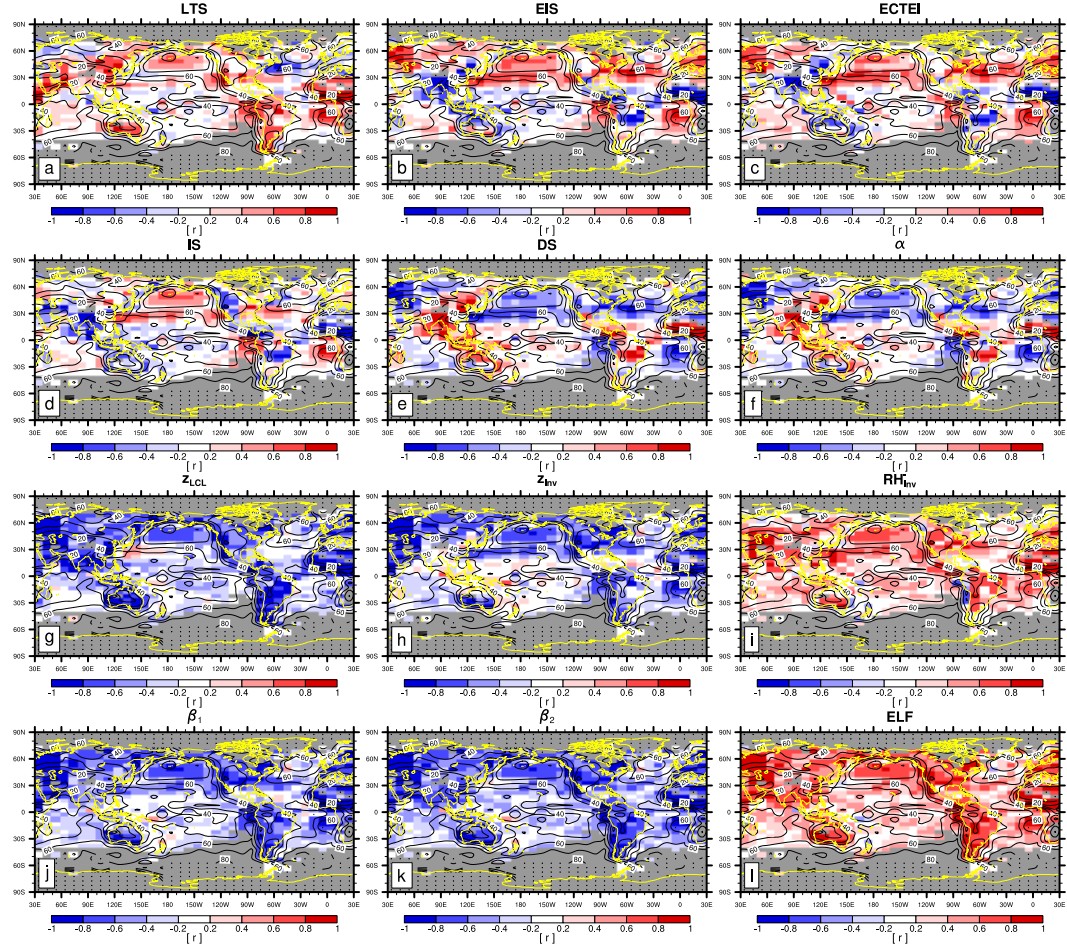

**Figure 9.** Global distribution of temporal (seasonal plus interannual) correlation coefficients between the seasonal LCA and various proxies, (a) LTS, (b) EIS, (c) ECTEI, (d) IS, (e) DS, (f) $\alpha$, (g) $z_{LCL}$, (h) $z_{inv}$, (i) $RH^{-}_{inv}$, (j) $\beta_1$, (k) $\beta_2$, and (l) ELF. The contour line denotes the climatological annual LCA reported by surface observers and the grid boxes with a climatological annual observation number below 100 are denoted by dots. Only the grid boxes with the number of effective seasonal data used for the correlation analysis equal to or larger than 50 out of the maximum 120 over the ocean and 72 over land are plotted. The grid boxes that are not satisfying these conditions are masked by dark grey color. The minimum magnitudes of correlation coefficients statistically significant at the 99.9% confidence level with the numbers of independent samples of 120 (ocean max), 72 (land max), and 50 (minimum requirement) are 0.3, 0.38, and 0.45, respectively. Only the data satisfying $0 < \alpha < 1$ are used for this figure.





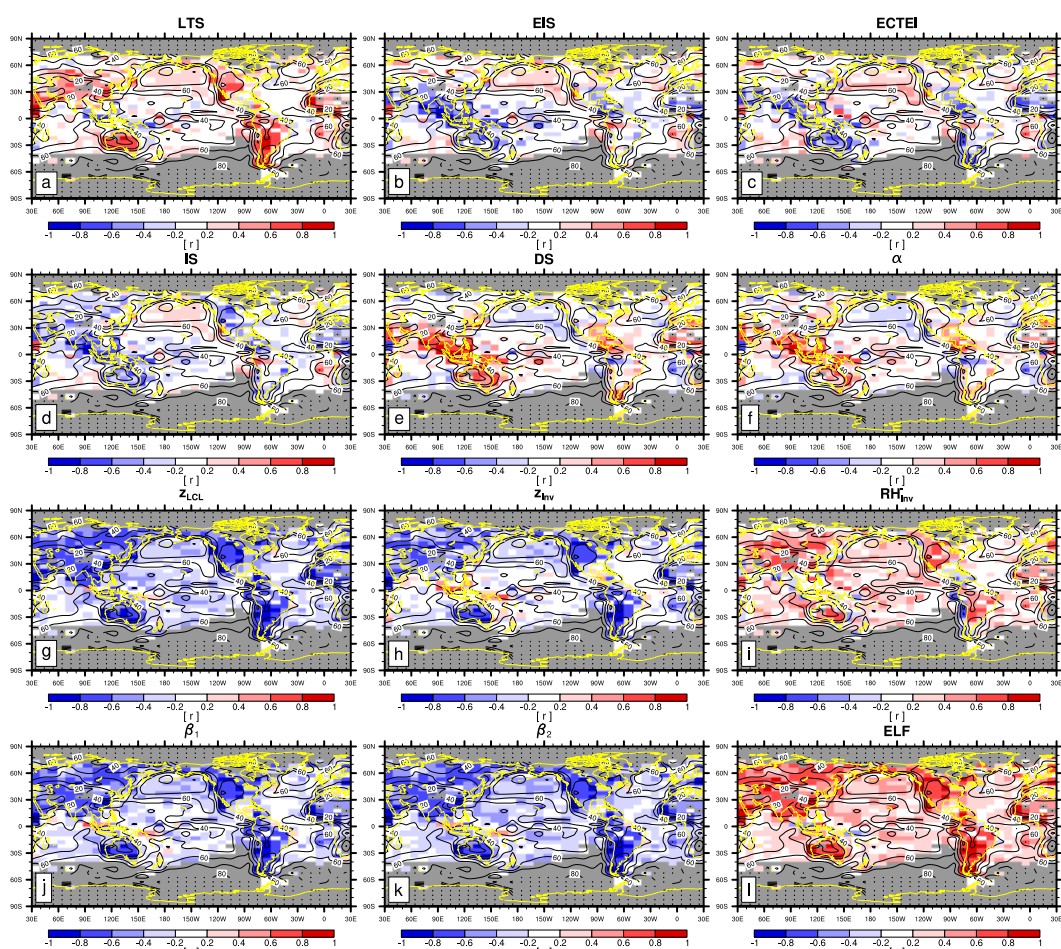

**Figure 10.** The same as Fig 9 but for interannual variation only without the seasonal cycle. This plot was generated by using the four seasonal data (DJF, MAM, JJA, SON) in each year with the seasonal climatology subtracted in each season, such that only interannual variation is included in this correlation analysis.





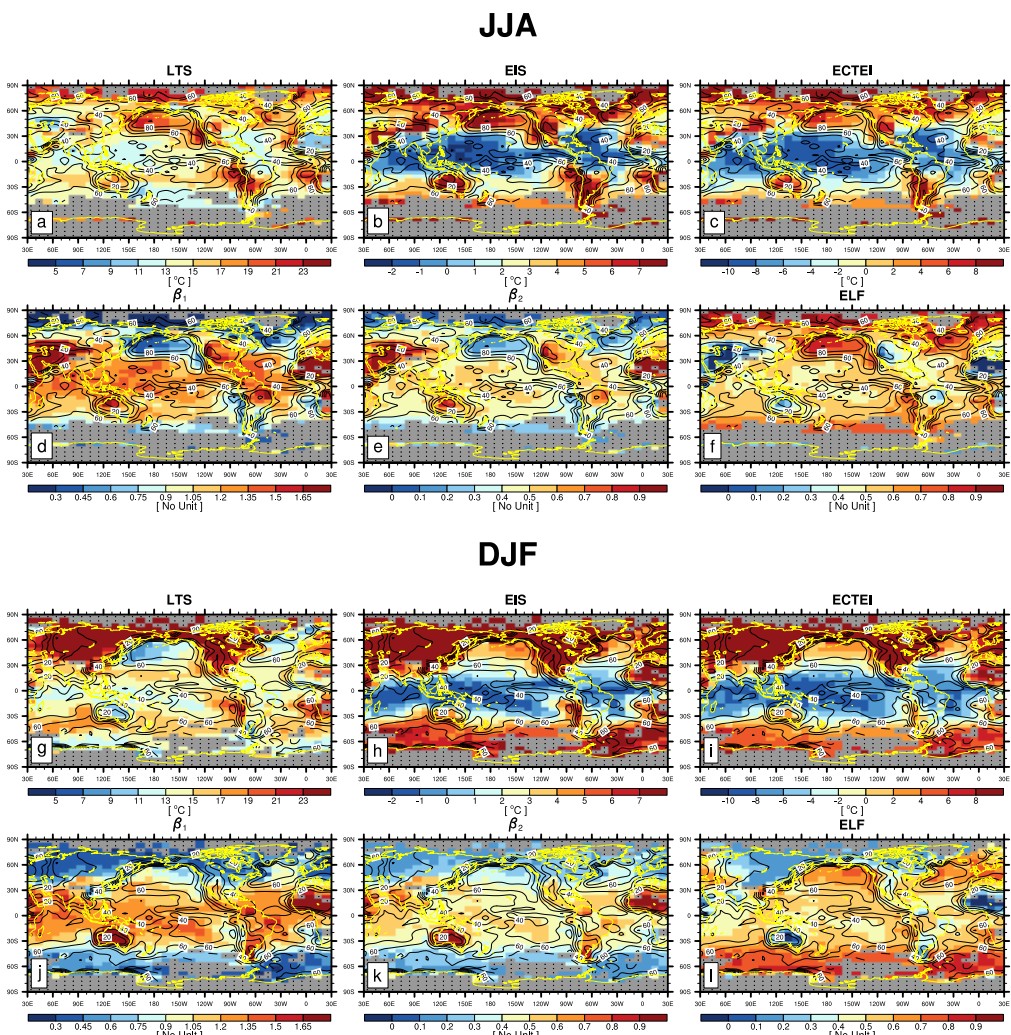

**Figure 11.** Global distribution of climatological LTS, EIS, ECTEI, $\beta_1$, $\beta_2$, and ELF during JJA (upper) and DJF (lower), respectively. The contour line denotes the climatological seasonal LCA reported by surface observers. These figures are the same as Figs 3 and 4 but were obtained from the analysis of all observation data (i.e., $0 < \alpha < 1$, $\alpha = 0$, and $\alpha = 1$).





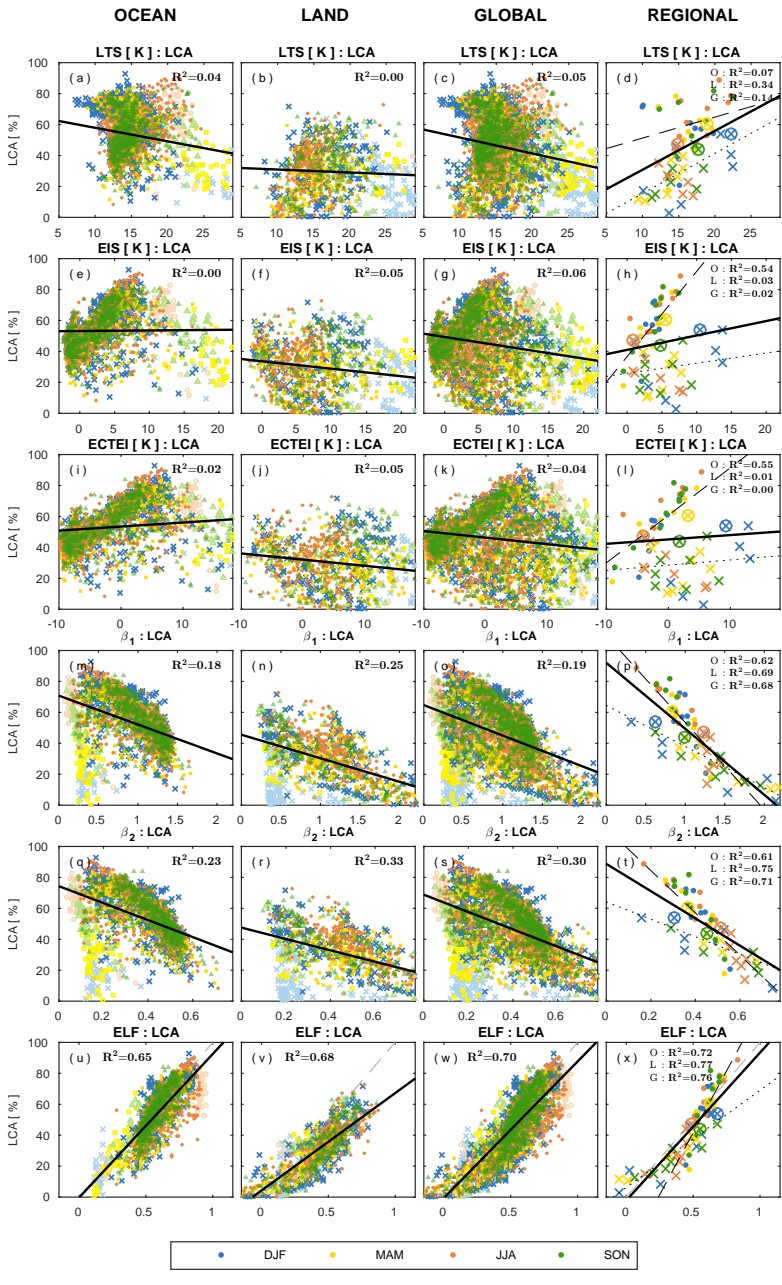

**Figure 12.** Scatter plots between LCA and six proxies over the (first column) ocean, (second) land, (third) globe, and (fourth) selected regions shown in Fig 7 with the least-square fitting lines and the fraction of variance ($R^2$) explained by the regression lines. The grey dashed line in the panels (u)-(x) denotes LCA = ELF. All observation data (i.e., $0 < \alpha < 1$, $\alpha = 0$, and $\alpha = 1$) are used for this figure. The grid data in the range of $0 \leq \alpha < 0.01$ (i.e., very stable regime) are denoted by light colors in the first three columns.





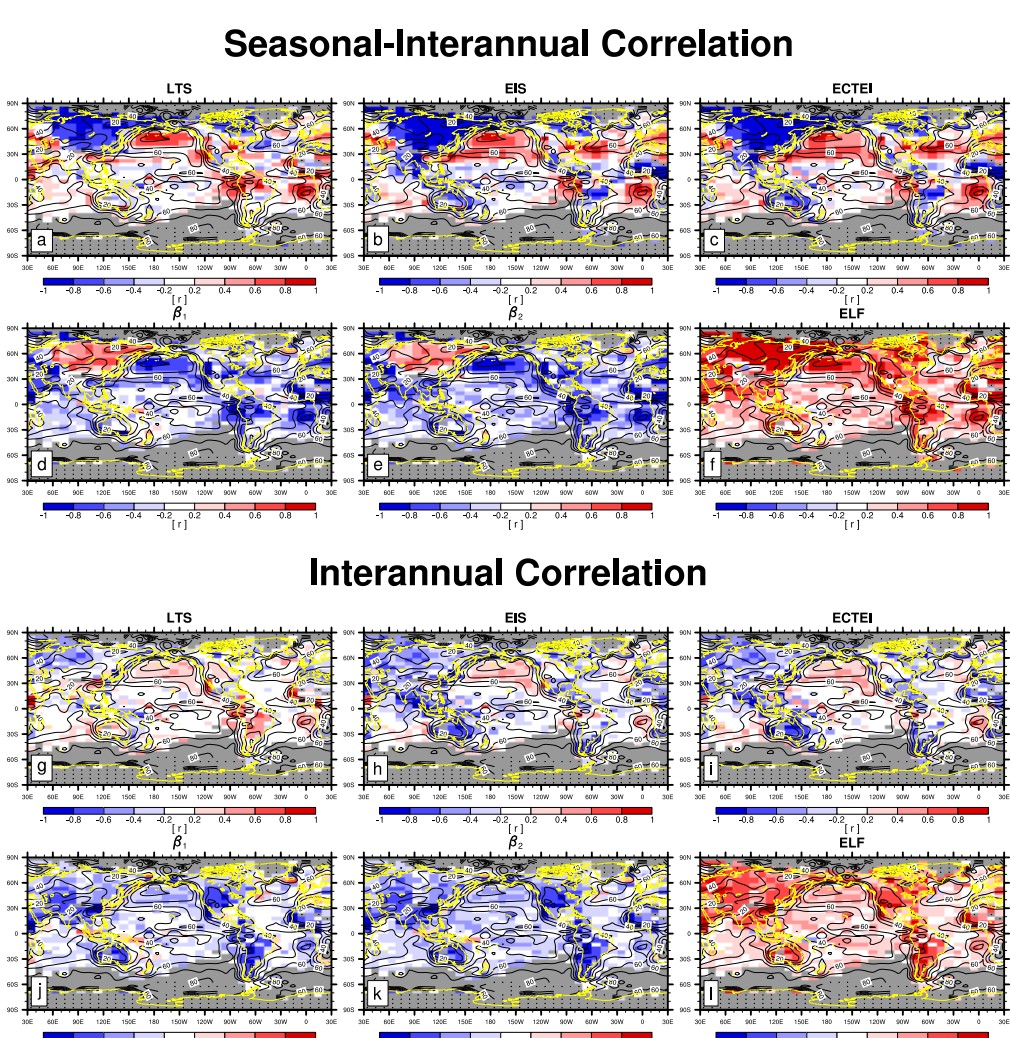

**Figure 13.** Global distribution of seasonal-interannual (upper) and interannual (lower) correlation coefficients between the seasonal LCA and six key proxies (LTS, EIS, ECTEI, $\beta_1$, $\beta_2$, and ELF). The contour line denotes the climatological annual LCA reported by surface observers. These figures are the same as Figs 9 and 10 but were obtained from the analysis of all observation data (i.e., $0 < \alpha < 1$, $\alpha = 0$, and $\alpha = 1$).



**Table 1.** Spatial-seasonal correlation coefficients between LCA and various proxies for the climatological seasonal data (DJF, MAM, JJA, SON) of each $5^o$latitude x $10^o$longitude grid box (values in the upper-right: over the ocean; values in the lower-left: over land; and values within the parenthesis: over the entire globe including the coast). Statistically significant correlations at the 99.9% confidence level from the Student t-test assuming independent samples are denoted by the bold characters. The same convention is applied to the following tables. Only the data satisfying $0 < \alpha < 1$ are used for this table.

| $r$ | LCA | LTS | EIS | ECTEI | IS | DS | $\alpha$ | $z_{LCL}$ | $z_{inv}$ | $RH^-_{inv}$ | $\beta_1$ | $\beta_2$ | ELF |
|---|---|---|---|---|---|---|---|---|---|---|---|---|---|
| **LCA** | 1 | -0.05 (0.17) | **0.62** (0.22) | **0.66** (0.19) | **0.39** (-0.09) | **-0.63** (-0.23) | **-0.58** (-0.17) | **-0.72** (-0.78) | **-0.69** (-0.67) | **0.62** (0.38) | **-0.75** (-0.80) | **-0.79** (-0.83) | **0.80** (0.83) |
| **LTS** | 0.18 | 1 | 0.07 (-0.08) | -0.17 (-0.31) | **0.57** (0.31) | 0.07 (0.23) | -0.09 (0.09) | 0.30 (-0.27) | 0.00 (-0.11) | -0.22 (-0.17) | 0.07 (-0.20) | 0.16 (-0.22) | -0.11 (0.26) |
| **EIS** | 0.07 | -0.13 | 1 | **0.96** (0.96) | **0.84** (0.82) | **-0.99** (-0.98) | **-0.99** (-0.98) | -0.38 (0.11) | **-0.96** (-0.73) | **0.75** (0.64) | **-0.90** (-0.44) | **-0.79** (-0.22) | **0.78** (0.22) |
| **ECTEI** | 0.07 | -0.35 | 0.96 | 1 | **0.68** (0.68) | **-0.98** (-0.98) | **-0.94** (-0.94) | -0.49 (0.15) | **-0.95** (-0.67) | **0.85** (0.72) | **-0.91** (-0.38) | **-0.84** (-0.17) | **0.83** (0.16) |
| **IS** | -0.39 | 0.21 | 0.71 | 0.59 | 1 | **-0.77** (-0.72) | **-0.86** (-0.82) | -0.05 (0.39) | **-0.76** (-0.41) | 0.46 (0.32) | **-0.66** (-0.10) | -0.49 (0.11) | 0.51 (-0.10) |
| **DS** | -0.16 | 0.35 | -0.95 | -0.97 | -0.50 | 1 | **0.98** (0.98) | 0.43 (-0.09) | **0.97** (0.75) | **-0.78** (-0.68) | **0.92** (0.46) | **0.82** (0.24) | **-0.81** (-0.23) |
| $\alpha$ | -0.11 | 0.17 | -0.97 | -0.94 | -0.66 | 0.96 | 1 | 0.37 (-0.12) | **0.97** (0.74) | **-0.74** (-0.65) | **0.91** (0.43) | **0.79** (0.21) | **-0.78** (-0.20) |
| $z_{LCL}$ | -0.83 | -0.40 | 0.15 | 0.21 | 0.53 | -0.09 | -0.11 | 1 | **0.59** (0.58) | **-0.60** (-0.27) | **0.73** (0.84) | **0.86** (0.94) | **-0.86** (-0.94) |
| $z_{inv}$ | -0.77 | -0.23 | -0.48 | -0.42 | 0.04 | 0.54 | 0.53 | 0.78 | 1 | **-0.81** (-0.71) | **0.98** (0.93) | **0.91** (0.82) | **-0.91** (-0.81) |
| $RH^-_{inv}$ | 0.71 | 0.22 | 0.27 | 0.29 | -0.18 | -0.34 | -0.30 | -0.69 | -0.78 | 1 | **-0.82** (-0.60) | **-0.81** (-0.48) | **0.80** (0.47) |
| $\beta_1$ | -0.85 | -0.33 | -0.20 | -0.13 | 0.29 | 0.26 | 0.25 | 0.93 | 0.95 | -0.78 | 1 | **0.97** (0.97) | **-0.97** (-0.97) |
| $\beta_2$ | -0.86 | -0.36 | -0.06 | 0.00 | 0.39 | 0.13 | 0.11 | 0.97 | 0.90 | -0.75 | 0.99 | 1 | **-0.99** (-1.00) |
| **ELF** | 0.86 | 0.39 | 0.06 | -0.01 | -0.38 | -0.11 | -0.10 | -0.97 | -0.89 | 0.75 | -0.99 | -1.00 | 1 |





**Table 2.** Temporal ($r_{s+i}$ : seasonal plus interannual, $r_i$ : interannual only) correlation coefficients of the seasonal LCA fitted to the individual proxy for selected regions shown in Figure 7. Only the data satisfying $0 < \alpha < 1$ are used for this table.

| Domain | | LTS | EIS | ECTEI | IS | DS | $\alpha$ | $z_{LCL}$ | $z_{inv}$ | $RH_{inv}^-$ | $\beta_1$ | $\beta_2$ | ELF |
|---|---|---|---|---|---|---|---|---|---|---|---|---|---|
| Peruvian | $r_{s+i}$ | 0.83 | 0.84 | 0.84 | 0.80 | -0.84 | -0.84 | -0.59 | -0.87 | 0.41 | -0.87 | -0.85 | 0.85 |
| | $r_i$ | 0.29 | 0.22 | 0.15 | 0.19 | -0.19 | -0.20 | -0.21 | -0.28 | 0.04 | -0.32 | -0.31 | 0.31 |
| Namibian | $r_{s+i}$ | 0.90 | 0.89 | 0.89 | 0.87 | -0.89 | -0.89 | -0.26 | -0.92 | 0.70 | -0.93 | -0.92 | 0.92 |
| | $r_i$ | 0.56 | 0.54 | 0.54 | 0.49 | -0.55 | -0.53 | -0.35 | -0.63 | 0.44 | -0.67 | -0.66 | 0.66 |
| Californian | $r_{s+i}$ | 0.70 | 0.71 | 0.61 | 0.70 | -0.67 | -0.69 | -0.35 | -0.72 | 0.67 | -0.72 | -0.67 | 0.67 |
| | $r_i$ | 0.18 | 0.30 | 0.33 | 0.25 | -0.31 | -0.29 | -0.05 | -0.30 | 0.22 | -0.28 | -0.22 | 0.22 |
| Australian | $r_{s+i}$ | 0.50 | 0.45 | 0.41 | 0.48 | -0.42 | -0.45 | -0.14 | -0.54 | 0.61 | -0.56 | -0.49 | 0.49 |
| | $r_i$ | 0.34 | 0.39 | 0.48 | 0.34 | -0.40 | -0.38 | -0.03 | -0.41 | 0.44 | -0.40 | -0.33 | 0.33 |
| Canarian | $r_{s+i}$ | 0.77 | 0.49 | 0.39 | 0.61 | -0.45 | -0.53 | -0.51 | -0.65 | 0.72 | -0.72 | -0.77 | 0.77 |
| | $r_i$ | 0.24 | 0.25 | 0.27 | 0.22 | -0.24 | -0.24 | 0.00 | -0.25 | 0.19 | -0.23 | -0.19 | 0.19 |
| Arabian | $r_{s+i}$ | 0.67 | -0.08 | -0.03 | 0.05 | 0.13 | 0.04 | -0.79 | -0.53 | 0.78 | -0.85 | -0.91 | 0.91 |
| | $r_i$ | -0.06 | -0.13 | -0.05 | -0.13 | 0.13 | 0.13 | -0.16 | 0.07 | 0.11 | -0.01 | -0.10 | 0.10 |
| North Pacific | $r_{s+i}$ | 0.75 | 0.83 | 0.69 | 0.78 | -0.74 | -0.79 | -0.84 | -0.84 | 0.81 | -0.86 | -0.87 | 0.87 |
| | $r_i$ | 0.29 | 0.26 | 0.19 | 0.26 | -0.16 | -0.23 | -0.40 | -0.36 | 0.21 | -0.42 | -0.44 | 0.44 |
| North Atlantic | $r_{s+i}$ | 0.38 | 0.39 | 0.29 | 0.39 | -0.34 | -0.40 | -0.62 | -0.50 | 0.38 | -0.55 | -0.59 | 0.59 |
| | $r_i$ | 0.42 | 0.32 | 0.28 | 0.36 | -0.28 | -0.35 | -0.64 | -0.50 | 0.31 | -0.58 | -0.63 | 0.63 |
| S.H. Circumpolar | $r_{s+i}$ | 0.27 | 0.07 | -0.25 | -0.29 | 0.16 | 0.24 | -0.76 | -0.51 | -0.21 | -0.65 | -0.71 | 0.72 |
| | $r_i$ | 0.24 | 0.04 | -0.25 | -0.44 | 0.14 | 0.26 | -0.86 | -0.61 | -0.28 | -0.76 | -0.83 | 0.82 |
| China | $r_{s+i}$ | 0.63 | 0.21 | 0.22 | 0.19 | -0.18 | -0.18 | -0.77 | -0.42 | 0.61 | -0.63 | -0.82 | 0.82 |
| | $r_i$ | 0.77 | 0.17 | 0.15 | 0.09 | -0.05 | -0.10 | -0.89 | -0.77 | 0.73 | -0.88 | -0.90 | 0.90 |
| India | $r_{s+i}$ | 0.67 | -0.85 | -0.85 | -0.75 | 0.91 | 0.72 | -0.85 | -0.57 | 0.84 | -0.74 | -0.79 | 0.79 |
| | $r_i$ | 0.40 | -0.39 | -0.28 | -0.40 | 0.44 | 0.37 | -0.62 | -0.45 | 0.66 | -0.57 | -0.59 | 0.60 |
| Europe | $r_{s+i}$ | -0.19 | 0.86 | 0.85 | 0.61 | -0.88 | -0.90 | -0.93 | -0.96 | 0.76 | -0.97 | -0.96 | 0.96 |
| | $r_i$ | 0.08 | 0.33 | 0.44 | -0.22 | -0.30 | -0.15 | -0.79 | -0.77 | 0.65 | -0.81 | -0.81 | 0.80 |
| Eastern US | $r_{s+i}$ | 0.57 | 0.87 | 0.83 | 0.81 | -0.86 | -0.88 | -0.69 | -0.96 | 0.90 | -0.94 | -0.90 | 0.90 |
| | $r_i$ | 0.66 | 0.31 | 0.18 | 0.22 | -0.15 | -0.30 | -0.74 | -0.81 | 0.62 | -0.82 | -0.80 | 0.80 |
| Western US | $r_{s+i}$ | 0.45 | -0.27 | -0.56 | -0.83 | -0.49 | 0.27 | -0.89 | -0.86 | 0.89 | -0.88 | -0.88 | 0.89 |
| | $r_i$ | 0.53 | -0.07 | -0.13 | -0.39 | -0.01 | 0.13 | -0.77 | -0.77 | 0.72 | -0.79 | -0.79 | 0.79 |
| South America | $r_{s+i}$ | 0.48 | -0.86 | -0.88 | -0.85 | 0.89 | 0.84 | -0.92 | -0.31 | 0.72 | -0.75 | -0.85 | 0.85 |
| | $r_i$ | 0.72 | -0.00 | -0.13 | -0.03 | 0.13 | -0.06 | -0.90 | -0.75 | 0.71 | -0.85 | -0.88 | 0.88 |
| Australia | $r_{s+i}$ | 0.82 | 0.38 | 0.07 | 0.25 | -0.13 | -0.38 | -0.91 | -0.77 | 0.81 | -0.84 | -0.86 | 0.86 |
| | $r_i$ | 0.88 | -0.32 | -0.64 | -0.38 | 0.73 | 0.38 | -0.93 | -0.87 | 0.86 | -0.91 | -0.92 | 0.92 |
| Southwest Sahara | $r_{s+i}$ | 0.97 | -0.89 | -0.92 | -0.83 | 0.95 | 0.82 | -0.97 | -0.94 | 0.98 | -0.98 | -0.98 | 0.98 |
| | $r_i$ | 0.74 | -0.44 | -0.52 | -0.27 | 0.63 | 0.26 | -0.83 | -0.73 | 0.65 | -0.82 | -0.84 | 0.84 |





**Table 3.** Combined spatial-seasonal-interannual correlation coefficients of the seasonal LCA fitted to the individual proxy with the same convention as Table 1. All seasonal data (DJF, MAM, JJA, SON) in each year in each $5^o$ latitude x $10^o$ longitude grid box are used. Only the data satisfying $0 < \alpha < 1$ are used for this table.

| $r$ | LCA | LTS | EIS | ECTEI | IS | DS | $\alpha$ | $z_{LCL}$ | $z_{inv}$ | $RH^-_{inv}$ | $\beta_1$ | $\beta_2$ | ELF |
|---|---|---|---|---|---|---|---|---|---|---|---|---|---|
| LCA | 1 | 0.02 | 0.53 | 0.58 | 0.34 | -0.54 | -0.49 | -0.60 | -0.60 | 0.57 | -0.66 | -0.70 | 0.70 |
| | | ( 0.17) | ( 0.20) | ( 0.19) | (-0.03) | (-0.21) | (-0.16) | (-0.70) | (-0.58) | ( 0.36) | (-0.72) | (-0.75) | ( 0.76) |
| LTS | 0.21 | 1 | 0.22 | -0.02 | 0.64 | -0.11 | -0.27 | 0.13 | -0.21 | -0.04 | -0.15 | -0.05 | 0.10 |
| | | | ( 0.06) | (-0.19) | ( 0.40) | ( 0.08) | (-0.07) | (-0.30) | (-0.25) | (-0.07) | (-0.31) | (-0.30) | ( 0.33) |
| EIS | 0.05 | -0.07 | 1 | 0.96 | 0.87 | -0.99 | -0.98 | -0.27 | -0.95 | 0.68 | -0.89 | -0.74 | 0.74 |
| | | | | ( 0.96) | ( 0.86) | (-0.99) | (-0.98) | ( 0.12) | (-0.78) | ( 0.63) | (-0.49) | (-0.25) | ( 0.24) |
| ECTEI | 0.06 | -0.31 | 0.95 | 1 | 0.71 | -0.97 | -0.92 | -0.36 | -0.93 | 0.80 | -0.89 | -0.78 | 0.77 |
| | | | | | ( 0.72) | (-0.97) | (-0.93) | ( 0.17) | (-0.71) | ( 0.73) | (-0.42) | (-0.19) | ( 0.18) |
| IS | -0.33 | 0.21 | 0.77 | 0.65 | 1 | -0.81 | -0.90 | -0.02 | -0.81 | 0.47 | -0.71 | -0.52 | 0.53 |
| | | | | | | (-0.78) | (-0.87) | ( 0.33) | (-0.55) | ( 0.39) | (-0.24) | ( 0.00) | ( 0.00) |
| DS | -0.11 | 0.26 | -0.96 | -0.97 | -0.61 | 1 | 0.98 | 0.30 | 0.96 | -0.71 | 0.90 | 0.77 | -0.75 |
| | | | | | | | ( 0.98) | (-0.12) | ( 0.78) | (-0.67) | ( 0.49) | ( 0.26) | (-0.24) |
| $\alpha$ | -0.07 | 0.10 | -0.97 | -0.93 | -0.75 | 0.96 | 1 | 0.25 | 0.97 | -0.68 | 0.89 | 0.74 | -0.73 |
| | | | | | | | | (-0.15) | ( 0.78) | (-0.63) | ( 0.48) | ( 0.24) | (-0.23) |
| $z_{LCL}$ | -0.77 | -0.46 | 0.19 | 0.27 | 0.51 | -0.16 | -0.18 | 1 | 0.49 | -0.50 | 0.66 | 0.83 | -0.83 |
| | | | | | | | | | ( 0.50) | (-0.21) | ( 0.80) | ( 0.93) | (-0.93) |
| $z_{inv}$ | -0.71 | -0.33 | -0.50 | -0.41 | -0.08 | 0.53 | 0.53 | 0.74 | 1 | -0.74 | 0.98 | 0.89 | -0.88 |
| | | | | | | | | | | (-0.69) | ( 0.92) | ( 0.79) | (-0.78) |
| $RH^-_{inv}$ | 0.62 | 0.18 | 0.35 | 0.39 | -0.04 | -0.40 | -0.36 | -0.56 | -0.73 | 1 | -0.76 | -0.74 | 0.73 |
| | | | | | | | | | | | (-0.57) | (-0.45) | ( 0.44) |
| $\beta_1$ | -0.80 | -0.42 | -0.19 | -0.10 | 0.21 | 0.22 | 0.22 | 0.92 | 0.94 | -0.69 | 1 | 0.96 | -0.96 |
| | | | | | | | | | | | | ( 0.96) | (-0.96) |
| $\beta_2$ | -0.81 | -0.44 | -0.03 | 0.05 | 0.33 | 0.07 | 0.06 | 0.97 | 0.87 | -0.64 | 0.99 | 1 | -0.99 |
| | | | | | | | | | | | | | (-1.00) |
| ELF | 0.80 | 0.47 | 0.03 | -0.06 | -0.33 | -0.06 | -0.05 | -0.97 | -0.87 | 0.64 | -0.98 | -1.00 | 1 |



**Table 4.** The same as Table 1, but all data (i.e., $0 < \alpha < 1$ and $\alpha = 0$ and $\alpha = 1$) are used for this table.

| $r$ | LCA | LTS | EIS | ECTEI | IS | DS | $\alpha$ | $z_{LCL}$ | $z_{inv}$ | $RH_{inv}^-$ | $\beta_1$ | $\beta_2$ | ELF |
|---|---|---|---|---|---|---|---|---|---|---|---|---|---|
| LCA | 1 | -0.20 | 0.01 | 0.13 | 0.51 | -0.41 | -0.31 | -0.54 | -0.38 | 0.44 | -0.43 | -0.48 | 0.81 |
| | | (-0.23) | (-0.23) | (-0.19) | ( 0.10) | (-0.01) | ( 0.06) | (-0.67) | (-0.26) | ( 0.19) | (-0.44) | (-0.54) | ( 0.84) |
| LTS | -0.06 | 1 | 0.79 | 0.69 | 0.23 | -0.49 | -0.61 | -0.27 | -0.59 | 0.37 | -0.56 | -0.50 | -0.16 |
| | | | ( 0.85) | ( 0.77) | ( 0.01) | (-0.54) | (-0.64) | (-0.21) | (-0.65) | ( 0.37) | (-0.58) | (-0.48) | (-0.15) |
| EIS | -0.21 | 0.90 | 1 | 0.98 | 0.20 | -0.84 | -0.88 | -0.50 | -0.87 | 0.58 | -0.85 | -0.81 | 0.02 |
| | | | | ( 0.99) | ( 0.03) | (-0.82) | (-0.87) | (-0.11) | (-0.81) | ( 0.49) | (-0.67) | (-0.52) | (-0.20) |
| ECTEI | -0.22 | 0.86 | 0.99 | 1 | 0.25 | -0.90 | -0.92 | -0.56 | -0.92 | 0.67 | -0.90 | -0.86 | 0.14 |
| | | | | | ( 0.10) | (-0.89) | (-0.91) | (-0.09) | (-0.84) | ( 0.56) | (-0.68) | (-0.52) | (-0.15) |
| IS | -0.14 | -0.36 | -0.36 | -0.29 | 1 | -0.59 | -0.57 | -0.10 | -0.52 | 0.52 | -0.47 | -0.39 | 0.66 |
| | | | | | | (-0.46) | (-0.44) | ( 0.36) | (-0.22) | ( 0.35) | (-0.04) | ( 0.10) | ( 0.23) |
| DS | 0.16 | -0.59 | -0.79 | -0.85 | -0.12 | 1 | 0.98 | 0.54 | 0.97 | -0.75 | 0.94 | 0.89 | -0.46 |
| | | | | | | | ( 0.98) | ( 0.04) | ( 0.87) | (-0.65) | ( 0.69) | ( 0.51) | (-0.10) |
| $\alpha$ | 0.15 | -0.65 | -0.82 | -0.87 | -0.10 | 0.99 | 1 | 0.54 | 0.99 | -0.74 | 0.96 | 0.90 | -0.40 |
| | | | | | | | | ( 0.04) | ( 0.89) | (-0.64) | ( 0.70) | ( 0.52) | (-0.05) |
| $z_{LCL}$ | -0.69 | -0.43 | -0.28 | -0.23 | 0.60 | 0.07 | 0.10 | 1 | 0.68 | -0.53 | 0.77 | 0.86 | -0.59 |
| | | | | | | | | | ( 0.49) | (-0.24) | ( 0.74) | ( 0.87) | (-0.73) |
| $z_{inv}$ | -0.32 | -0.74 | -0.77 | -0.78 | 0.30 | 0.76 | 0.79 | 0.69 | 1 | -0.76 | 0.99 | 0.96 | -0.47 |
| | | | | | | | | | | (-0.67) | ( 0.95) | ( 0.85) | (-0.38) |
| $RH_{inv}^-$ | 0.54 | 0.26 | 0.13 | 0.15 | -0.11 | -0.17 | -0.19 | -0.62 | -0.52 | 1 | -0.76 | -0.74 | 0.49 |
| | | | | | | | | | | | (-0.61) | (-0.52) | ( 0.32) |
| $\beta_1$ | -0.50 | -0.67 | -0.63 | -0.61 | 0.45 | 0.53 | 0.57 | 0.88 | 0.95 | -0.61 | 1 | 0.99 | -0.51 |
| | | | | | | | | | | | | ( 0.97) | (-0.55) |
| $\beta_2$ | -0.58 | -0.62 | -0.53 | -0.51 | 0.51 | 0.40 | 0.44 | 0.94 | 0.90 | -0.62 | 0.99 | 1 | -0.55 |
| | | | | | | | | | | | | | (-0.65) |
| ELF | 0.82 | 0.03 | -0.16 | -0.16 | -0.03 | 0.01 | -0.00 | -0.76 | -0.47 | 0.66 | -0.63 | -0.69 | 1 |





**Table 5.** The same as Table 2, but all data (i.e., $0 < \alpha < 1$ and $\alpha = 0$ and $\alpha = 1$) are used for this table.

| Domain | | LTS | EIS | ECTEI | IS | DS | $\alpha$ | $z_{\mathrm{LCL}}$ | $z_{\mathrm{inv}}$ | $RH^-_{\mathrm{inv}}$ | $\beta_1$ | $\beta_2$ | ELF |
|---|---|---|---|---|---|---|---|---|---|---|---|---|---|
| Peruvian | $r_{s+i}$ | **0.85** | **0.86** | **0.87** | **0.83** | **-0.86** | **-0.86** | -0.30 | **-0.88** | **0.44** | **-0.87** | **-0.84** | **0.84** |
| | $r_i$ | 0.30 | 0.27 | 0.24 | 0.25 | -0.26 | -0.26 | -0.05 | -0.30 | 0.07 | -0.29 | -0.24 | 0.24 |
| Namibian | $r_{s+i}$ | **0.90** | **0.89** | **0.89** | **0.88** | **-0.89** | **-0.89** | -0.16 | **-0.92** | **0.70** | **-0.93** | **-0.91** | **0.91** |
| | $r_i$ | **0.54** | **0.52** | **0.52** | **0.50** | **-0.55** | **-0.53** | -0.27 | **-0.61** | **0.45** | **-0.65** | **-0.64** | **0.64** |
| Californian | $r_{s+i}$ | **0.72** | **0.72** | **0.65** | **0.71** | **-0.68** | **-0.69** | -0.32 | **-0.72** | **0.67** | **-0.72** | **-0.69** | **0.70** |
| | $r_i$ | 0.18 | 0.28 | **0.33** | 0.24 | **-0.31** | -0.29 | 0.03 | -0.28 | 0.22 | -0.25 | -0.21 | 0.21 |
| Australian | $r_{s+i}$ | **0.53** | **0.49** | **0.46** | **0.52** | **-0.46** | **-0.49** | -0.07 | **-0.57** | **0.63** | **-0.58** | **-0.49** | **0.49** |
| | $r_i$ | **0.37** | **0.42** | **0.50** | **0.38** | **-0.42** | **-0.41** | -0.00 | **-0.44** | **0.47** | **-0.42** | **-0.35** | **0.35** |
| Canarian | $r_{s+i}$ | **0.77** | **0.51** | **0.41** | **0.62** | **-0.47** | **-0.55** | **-0.49** | **-0.66** | **0.71** | **-0.73** | **-0.77** | **0.77** |
| | $r_i$ | 0.22 | 0.22 | 0.25 | 0.19 | -0.23 | -0.21 | -0.01 | -0.23 | 0.15 | -0.22 | -0.20 | 0.20 |
| Arabian | $r_{s+i}$ | **0.66** | -0.02 | 0.01 | 0.10 | 0.12 | -0.02 | **-0.78** | **-0.54** | **0.77** | **-0.85** | **-0.91** | **0.91** |
| | $r_i$ | -0.05 | -0.13 | -0.07 | -0.13 | 0.13 | 0.12 | -0.18 | 0.06 | 0.08 | -0.01 | -0.12 | 0.11 |
| North Pacific | $r_{s+i}$ | **0.86** | **0.84** | **0.79** | **0.85** | **-0.79** | **-0.83** | **-0.84** | **-0.87** | **0.78** | **-0.90** | **-0.91** | **0.91** |
| | $r_i$ | **0.37** | **0.36** | **0.35** | **0.35** | **-0.34** | **-0.35** | -0.26 | **-0.41** | 0.30 | **-0.45** | **-0.45** | **0.45** |
| North Atlantic | $r_{s+i}$ | **0.46** | **0.43** | **0.35** | **0.44** | **-0.40** | **-0.45** | **-0.66** | **-0.54** | **0.34** | **-0.59** | **-0.65** | **0.64** |
| | $r_i$ | **0.43** | **0.37** | **0.34** | **0.37** | **-0.36** | **-0.38** | **-0.64** | **-0.51** | 0.22 | **-0.59** | **-0.67** | **0.67** |
| S.H. Circumpolar | $r_{s+i}$ | **-0.31** | **-0.45** | **-0.48** | -0.14 | **0.45** | **0.48** | **-0.60** | 0.19 | **-0.36** | -0.05 | -0.28 | **0.57** |
| | $r_i$ | **-0.30** | **-0.38** | **-0.42** | **-0.38** | **0.40** | **0.46** | **-0.75** | 0.13 | **-0.54** | -0.15 | **-0.43** | **0.56** |
| China | $r_{s+i}$ | **0.47** | 0.35 | 0.36 | 0.32 | -0.33 | -0.33 | **-0.88** | **-0.48** | 0.15 | **-0.60** | **-0.71** | **0.71** |
| | $r_i$ | **0.53** | 0.14 | 0.16 | -0.05 | 0.05 | 0.01 | **-0.91** | **-0.75** | **0.48** | **-0.90** | **-0.93** | **0.92** |
| India | $r_{s+i}$ | 0.30 | **-0.81** | **-0.86** | **-0.78** | **0.92** | **0.76** | **-0.82** | -0.30 | **0.82** | **-0.61** | **-0.70** | **0.70** |
| | $r_i$ | -0.01 | **-0.48** | **-0.41** | **-0.47** | **0.52** | **0.47** | **-0.58** | -0.14 | **0.57** | **-0.41** | **-0.47** | **0.48** |
| Europe | $r_{s+i}$ | **0.76** | **0.73** | **0.73** | 0.38 | **-0.74** | **-0.76** | **-0.91** | **-0.84** | -0.16 | **-0.87** | **-0.89** | **0.85** |
| | $r_i$ | **-0.48** | **-0.56** | **-0.53** | 0.34 | 0.22 | 0.25 | **-0.57** | -0.22 | **0.54** | **-0.39** | **-0.46** | **0.82** |
| Eastern US | $r_{s+i}$ | **0.78** | **0.78** | **0.78** | 0.45 | **-0.74** | **-0.75** | **-0.54** | **-0.82** | **0.57** | **-0.85** | **-0.86** | **0.77** |
| | $r_i$ | **0.44** | 0.11 | 0.09 | -0.02 | 0.04 | -0.08 | **-0.82** | **-0.69** | **0.61** | **-0.82** | **-0.84** | **0.79** |
| Western US | $r_{s+i}$ | **0.62** | **0.66** | **0.69** | **-0.91** | **-0.71** | **-0.61** | **-0.91** | **-0.87** | **0.82** | **-0.89** | **-0.89** | **0.90** |
| | $r_i$ | 0.25 | 0.12 | 0.13 | **-0.53** | -0.07 | -0.00 | **-0.78** | **-0.71** | **0.62** | **-0.75** | **-0.75** | **0.73** |
| South America | $r_{s+i}$ | -0.18 | **-0.84** | **-0.88** | **-0.84** | **0.88** | **0.83** | **-0.90** | 0.17 | **0.80** | **-0.49** | **-0.72** | **0.72** |
| | $r_i$ | **0.48** | -0.15 | -0.20 | -0.11 | 0.25 | 0.08 | **-0.89** | **-0.59** | **0.72** | **-0.78** | **-0.83** | **0.83** |
| Australia | $r_{s+i}$ | -0.12 | **-0.43** | **-0.53** | **-0.50** | **0.57** | **0.46** | -0.37 | 0.09 | 0.22 | -0.07 | -0.13 | 0.13 |
| | $r_i$ | **0.42** | **-0.74** | **-0.86** | **-0.55** | **0.86** | **0.70** | **-0.92** | **-0.67** | **0.83** | **-0.84** | **-0.86** | **0.87** |
| Southwest Sahara | $r_{s+i}$ | 0.10 | **-0.87** | **-0.91** | **-0.87** | **0.94** | **0.86** | **-0.98** | **-0.62** | **0.94** | **-0.96** | **-0.97** | **0.97** |
| | $r_i$ | **0.45** | -0.38 | **-0.45** | -0.34 | **0.57** | 0.32 | **-0.83** | **-0.65** | **0.65** | **-0.83** | **-0.85** | **0.86** |





**Table 6.** The same as Table 3, but all data (i.e., $0 < \alpha < 1$ and $\alpha = 0$ and $\alpha = 1$) are used for this table.

| $r$ | LCA | LTS | EIS | ECTEI | IS | DS | $\alpha$ | $z_{LCL}$ | $z_{inv}$ | $RH_{inv}^-$ | $\beta_1$ | $\beta_2$ | ELF |
|---|---|---|---|---|---|---|---|---|---|---|---|---|---|
| LCA | 1 | -0.03 | 0.16 | 0.26 | 0.41 | -0.42 | -0.35 | -0.50 | -0.42 | 0.44 | -0.47 | -0.51 | 0.71 |
|  |  | (-0.15) | (-0.19) | (-0.14) | ( 0.09) | (-0.03) | ( 0.03) | (-0.63) | (-0.24) | ( 0.20) | (-0.41) | (-0.52) | ( 0.77) |
| LTS | -0.07 | 1 | 0.71 | 0.59 | 0.51 | -0.49 | -0.62 | -0.20 | -0.61 | 0.40 | -0.57 | -0.50 | 0.11 |
|  |  |  | ( 0.81) | ( 0.71) | ( 0.25) | (-0.54) | (-0.65) | (-0.17) | (-0.66) | ( 0.40) | (-0.59) | (-0.49) | (-0.03) |
| EIS | -0.22 | 0.89 | 1 | 0.98 | 0.44 | -0.87 | -0.90 | -0.38 | -0.90 | 0.58 | -0.86 | -0.79 | 0.24 |
|  |  |  |  | ( 0.98) | ( 0.23) | (-0.84) | (-0.88) | (-0.04) | (-0.82) | ( 0.51) | (-0.68) | (-0.51) | (-0.12) |
| ECTEI | -0.22 | 0.83 | 0.99 | 1 | 0.46 | -0.92 | -0.92 | -0.43 | -0.92 | 0.65 | -0.90 | -0.84 | 0.34 |
|  |  |  |  |  | ( 0.29) | (-0.90) | (-0.91) | (-0.01) | (-0.84) | ( 0.59) | (-0.68) | (-0.51) | (-0.08) |
| IS | -0.11 | -0.23 | -0.23 | -0.15 | 1 | -0.71 | -0.73 | -0.07 | -0.67 | 0.56 | -0.61 | -0.49 | 0.62 |
|  |  |  |  |  |  | (-0.60) | (-0.60) | ( 0.30) | (-0.42) | ( 0.45) | (-0.23) | (-0.05) | ( 0.25) |
| DS | 0.15 | -0.59 | -0.80 | -0.86 | -0.26 | 1 | 0.97 | 0.38 | 0.96 | -0.67 | 0.92 | 0.84 | -0.56 |
|  |  |  |  |  |  |  | ( 0.98) | (-0.03) | ( 0.88) | (-0.64) | ( 0.70) | ( 0.50) | (-0.15) |
| $\alpha$ | 0.15 | -0.65 | -0.83 | -0.88 | -0.24 | 0.99 | 1 | 0.37 | 0.98 | -0.69 | 0.94 | 0.85 | -0.52 |
|  |  |  |  |  |  |  |  | (-0.03) | ( 0.90) | (-0.65) | ( 0.72) | ( 0.51) | (-0.12) |
| $z_{LCL}$ | -0.67 | -0.37 | -0.19 | -0.14 | 0.54 | -0.00 | 0.02 | 1 | 0.55 | -0.43 | 0.67 | 0.81 | -0.67 |
|  |  |  |  |  |  |  |  |  | ( 0.40) | (-0.18) | ( 0.68) | ( 0.84) | (-0.75) |
| $z_{inv}$ | -0.30 | -0.73 | -0.76 | -0.76 | 0.15 | 0.76 | 0.79 | 0.63 | 1 | -0.71 | 0.99 | 0.93 | -0.61 |
|  |  |  |  |  |  |  |  |  |  | (-0.67) | ( 0.95) | ( 0.83) | (-0.43) |
| $RH_{inv}^-$ | 0.42 | 0.30 | 0.22 | 0.27 | 0.04 | -0.32 | -0.32 | -0.48 | -0.54 | 1 | -0.71 | -0.69 | 0.54 |
|  |  |  |  |  |  |  |  |  |  |  | (-0.60) | (-0.51) | ( 0.33) |
| $\beta_1$ | -0.48 | -0.66 | -0.60 | -0.58 | 0.33 | 0.52 | 0.55 | 0.85 | 0.95 | -0.57 | 1 | 0.98 | -0.67 |
|  |  |  |  |  |  |  |  |  |  |  |  | ( 0.96) | (-0.61) |
| $\beta_2$ | -0.57 | -0.59 | -0.48 | -0.45 | 0.42 | 0.36 | 0.39 | 0.93 | 0.87 | -0.55 | 0.98 | 1 | -0.71 |
|  |  |  |  |  |  |  |  |  |  |  |  |  | (-0.71) |
| ELF | 0.79 | 0.02 | -0.18 | -0.19 | -0.01 | 0.03 | 0.02 | -0.78 | -0.46 | 0.52 | -0.64 | -0.71 | 1 |