# Peer review of "Heuristic Estimation of Low-Level Cloud Fraction over the Globe Based on a Decoupling Parameterization"

_Atmospheric Chemistry and Physics, 2019_

## Referee Comment (RC1) · Hideaki Kawai (Referee) · 26 Feb 2019

Hideaki Kawai

Meteorological Research Institute, JMA

The topic of this manuscript is very interesting and important.

The authors successfully developed a powerful proxy for global LCA that is applicable not only over the ocean but also over land. The results are striking and beneficial for a lot of readers, although the proxy is empirical. The manuscript is well-written and well-organized. I appreciate that the authors squarely show all the results of 12 proxies and their cross-correlations. It should be also appreciated that the authors examined not only special-seasonal correlation but also temporal (seasonal plus interannual) regressions.

I basically recommend publication of the manuscript after the authors make minor revisions.

(I have reviewed this manuscript in the past. The authors already addressed most of my concerns and revised the manuscript properly.) I still would like to suggest only two things, which can be easily done. (Although the authors say that they will submit another paper which includes the following analyses, I guess they are important to clarify the contents of this manuscript and to avoid confusion for the readers. But it depends on the editor's and other reviewers' judgements.)

**Suggestions:**

(A) Analysis using all samples over the ocean (Fig. 12 and related tables):

For example, the EIS range in the panel of EIS for Ocean in Fig. 12 is very wide. I guess data with extremely large EIS and small LCA are from sea ice regions. I understand that one of the advantages of the authors' new proxy is that it can be used not only over the ice-free ocean but also sea ice regions (and over land, of course). However, the relationships between LCA and the indices are usually supposed to be used only over the ice-free ocean. Therefore, I guess readers want to see the comparison of their relationships only over the ice-free ocean (using all samples). In addition, readers will be confused by the very low correlation for EIS etc. in Fig. 12 and related tables. Can you eliminate data over sea-ice from the 'OCEAN' panels (and modify tables), if they are included?

(B) Analysis using only stratiform clouds:

In many studies related to low clouds and the proxies, the definition of low clouds is stratiform clouds that include stratocumulus, stratus, and fog. But, in the authors' study, low clouds include cumulus too. I think it is one of the advantages of this study and it is good. However, I guess many readers want to see the results using the same definition of low clouds that has been conventionally used, especially over the ocean (only for the ice-free ocean). So, could you add the figure and tables as supporting information, and give short discussion about the difference between including and not including cumulus (the authors don't need to analyze the relationships for individual low-level cloud types, just for stratiform clouds which have the same definition in previous studies)?

---

## Referee Comment (RC2) · Timothy Myers (Referee) · 5 Mar 2019

In the present work, Park and Shin propose new proxies for the analysis of variations in low-level cloud fraction, motivated by the previously developed and widely used metrics of lower tropospheric stability (LTS) and estimated inversion strength (EIS). Their proxies, including two so-called low cloud suppression parameters (LCS) and estimated low cloud fraction (ELF), are related to inversion height and the lifting condensation level of surface air. Remarkably, these parameters explain most of the combined spatial-seasonal-interannual variability of low-level cloud fraction over both land and ocean globally, which is much larger than the fraction of variability explained by either LTS or

[Figure]

EIS. It is also quite fascinating that in the framework developed by the authors, EIS is found to be more related to decoupling strength than it is to inversion strength. The results will surely be of interest to the community of researchers studying low clouds. This is especially true since the proxies the authors develop have firm theoretical underpinnings, high relative skill, and general applicability to multiple regions and surface types.

I have reviewed the paper before for another journal, and the authors have addressed all of my previous comments. The paper can be published almost as is, though there are two important typos that need to be corrected first. In the Abstract and Implication section, I am certain that the authors mean to write "(or LCS)" after "ELF", instead of "(or LTS)".

-Tim Myers

---

## Author Comment (AC1) · 3 Apr 2019

**Supplement of "Heuristic Estimation of Low-Level Cloud Fraction over the Globe Based on a Decoupling Parameterization"**

Sungsu Park[1] and Jihoon Shin[1]

[1]School of Earth and Environmental Sciences, Seoul National University, Seoul, South Korea

**Correspondence:** Sungsu Park (sungsup@snu.ac.kr)

**Supplement 1 : Analysis with Stratiform Clouds Only between 60$^o$S and 60$^o$N**

Table S1 and Figure S1 are the same analysis as Table 3 and Figure 12, except that only stratiform LCA (the sum of stratocumulus, stratus, and fog) between 60$^o$S and 60$^o$N over the ice-free ocean are used instead of all LCA over the entire globe. Over the ocean, the correlations between the proxies and stratiform LCA tend to be higher than the correlations with entire LCA (Table S1). The correlations of LTS/EIS/ECTEI with LCA show relatively larger increases ($\Delta r = 0.14 \sim 0.3$) than those of $\beta_1/\beta_2$/ELF ($\Delta r = 0.08 \sim 0.12$), because LTS/EIS/ECTEI are designed to be applicable mainly over the marine stratiform cloud regions. The improved performance of EIS/ECTEI over the ocean with stratiform LCA are also evident in Figure S1 (compare with Figure 12). Note that not only EIS/ECTEI but also $\beta_1/\beta_2$/ELF shows improved performance. Overall, ELF shows the best performance in diagnosing stratform LCA.

**Table S1.** Combined spatial-seasonal-interannual correlation coefficients of the seasonal stratiform LCA (defined as the fraction of stratocumulus, stratus, and fog) fitted to the individual proxy. All seasonal data (DJF, MAM, JJA, SON) in each year in each $5^o$latitude x $10^o$longitude grid box are used. All observation data (i.e., $0 < \alpha < 1$ and $\alpha = 0$ and $\alpha = 1$) between $60^\circ$S and $60^\circ$N are used for this table.

| $r$ | LCA | LTS | EIS | ECTEI | IS | DS | $\alpha$ | $z_{LCL}$ | $z_{inv}$ | $RH_{inv}^-$ | $\beta_1$ | $\beta_2$ | ELF |
|---|---|---|---|---|---|---|---|---|---|---|---|---|---|
| LCA | 1 | 0.32 | 0.69 | 0.73 | 0.56 | -0.72 | -0.67 | -0.52 | -0.75 | 0.63 | -0.78 | -0.78 | 0.80 |
|  |  | ( 0.21) | ( 0.26) | ( 0.31) | ( 0.30) | (-0.42) | (-0.35) | (-0.51) | (-0.61) | ( 0.45) | (-0.70) | (-0.69) | ( 0.73) |
| LTS | 0.27 | 1 | 0.56 | 0.39 | 0.82 | -0.47 | -0.64 | -0.06 | -0.61 | 0.45 | -0.56 | -0.44 | 0.44 |
|  |  |  | ( 0.70) | ( 0.58) | ( 0.57) | (-0.54) | (-0.67) | (-0.08) | (-0.70) | ( 0.45) | (-0.58) | (-0.43) | ( 0.23) |
| EIS | 0.03 | 0.83 | 1 | 0.97 | 0.84 | -0.98 | -0.98 | -0.22 | -0.96 | 0.64 | -0.91 | -0.79 | 0.73 |
|  |  |  |  | ( 0.98) | ( 0.61) | (-0.91) | (-0.94) | ( 0.18) | (-0.81) | ( 0.54) | (-0.57) | (-0.32) | ( 0.06) |
| ECTEI | 0.01 | 0.76 | 0.99 | 1 | 0.74 | -0.97 | -0.94 | -0.28 | -0.94 | 0.69 | -0.90 | -0.81 | 0.75 |
|  |  |  |  |  | ( 0.61) | (-0.94) | (-0.93) | ( 0.21) | (-0.80) | ( 0.60) | (-0.54) | (-0.30) | ( 0.06) |
| IS | -0.06 | 0.08 | 0.15 | 0.23 | 1 | -0.84 | -0.91 | -0.02 | -0.85 | 0.59 | -0.77 | -0.60 | 0.61 |
|  |  |  |  |  |  | (-0.78) | (-0.82) | ( 0.35) | (-0.61) | ( 0.50) | (-0.34) | (-0.08) | ( 0.11) |
| DS | -0.05 | -0.62 | -0.87 | -0.91 | -0.52 | 1 | 0.97 | 0.22 | 0.95 | -0.63 | 0.90 | 0.79 | -0.75 |
|  |  |  |  |  |  |  | ( 0.97) | (-0.19) | ( 0.85) | (-0.59) | ( 0.59) | ( 0.33) | (-0.19) |
| $\alpha$ | -0.06 | -0.69 | -0.89 | -0.92 | -0.52 | 0.98 | 1 | 0.19 | 0.98 | -0.66 | 0.92 | 0.78 | -0.74 |
|  |  |  |  |  |  |  |  | (-0.21) | ( 0.86) | (-0.61) | ( 0.59) | ( 0.32) | (-0.16) |
| $z_{LCL}$ | -0.66 | -0.33 | 0.01 | 0.09 | 0.55 | -0.20 | -0.18 | 1 | 0.40 | -0.36 | 0.57 | 0.76 | -0.76 |
|  |  |  |  |  |  |  |  |  | ( 0.31) | (-0.12) | ( 0.66) | ( 0.85) | (-0.86) |
| $z_{inv}$ | -0.51 | -0.82 | -0.75 | -0.72 | -0.06 | 0.70 | 0.73 | 0.55 | 1 | -0.70 | 0.98 | 0.90 | -0.86 |
|  |  |  |  |  |  |  |  |  |  | (-0.65) | ( 0.92) | ( 0.76) | (-0.61) |
| $RH_{inv}^-$ | 0.43 | 0.40 | 0.24 | 0.28 | 0.08 | -0.28 | -0.30 | -0.46 | -0.57 | 1 | -0.70 | -0.67 | 0.66 |
|  |  |  |  |  |  |  |  |  |  |  | (-0.56) | (-0.44) | ( 0.37) |
| $\beta_1$ | -0.65 | -0.70 | -0.49 | -0.44 | 0.22 | 0.37 | 0.40 | 0.83 | 0.92 | -0.59 | 1 | 0.96 | -0.93 |
|  |  |  |  |  |  |  |  |  |  |  |  | ( 0.95) | (-0.83) |
| $\beta_2$ | -0.68 | -0.60 | -0.33 | -0.26 | 0.36 | 0.17 | 0.21 | 0.93 | 0.82 | -0.56 | 0.98 | 1 | -0.98 |
|  |  |  |  |  |  |  |  |  |  |  |  |  | (-0.92) |
| ELF | 0.76 | 0.27 | -0.09 | -0.13 | -0.18 | 0.07 | 0.05 | -0.88 | -0.58 | 0.54 | -0.80 | -0.86 | 1 |

[Figure]

**Figure S1.** Scatter plots between stratiform LCA (defined as the fraction of stratocumulus, stratus, and fog) and six proxies (first column) ocean, (second) land, (third) globe, and (fourth) selected regions shown in Fig 7 with the least-square fitting lines and the fraction of variance ($R^2$) explained by the regression lines. The grey dashed line in the panels (u)-(x) denotes LCA = ELF. All observation data (i.e., $0 < \alpha < 1$, $\alpha = 0$, and $\alpha = 1$) between 60°S and 60°N are used for this figure. The grid data in the range of $0 \le \alpha < 0.01$ (i.e., very stable regime) are denoted by light colors in the first three columns.

---

## Author Comment (AC2) · 3 Apr 2019

Dear Dr. Myers

Thank you very much for the very careful comment.

Following your comment, we corrected LTS to LCS in L.18 (page.1) and L.14 (p.16) in the revised draft.

Sungsu

2019.